# Unbiased Learning of Deep Generative Models with Structured Discrete Representations

**Harry Bendekgey, Gabriel Hope, Erik B. Sudderth**
{hbendekg, hopej, sudderth}@uci.edu
Department of Computer Science, University of California, Irvine

## Abstract

By composing graphical models with deep learning architectures, we learn generative models with the strengths of both frameworks. The structured variational autoencoder (SVAE) inherits structure and interpretability from graphical models, and flexible likelihoods for high-dimensional data from deep learning, but poses substantial optimization challenges. We propose novel algorithms for learning SVAEs, and are the first to demonstrate the SVAE's ability to handle multimodal uncertainty when data is missing by incorporating discrete latent variables. Our memory-efficient implicit differentiation scheme makes the SVAE tractable to learn via gradient descent, while demonstrating robustness to incomplete optimization. To more rapidly learn accurate graphical model parameters, we derive a method for computing natural gradients without manual derivations, which avoids biases found in prior work. These optimization innovations enable the first comparisons of the SVAE to state-of-the-art time series models, where the SVAE performs competitively while learning interpretable and structured discrete data representations.

## 1 Introduction

Advances in deep learning have dramatically increased the expressivity of machine learning models at great cost to their interpretability. This trade-off can be seen in deep generative models that produce remarkably accurate synthetic data, but often fail to illuminate the data's underlying factors of variation, and cannot easily incorporate domain knowledge. The *structured variational autoencoder* (SVAE, Johnson et al. [29]) aims to elegantly address these issues by combining probabilistic graphical models [62] with the VAE [33], gaining both flexibility and interpretability. But since its 2016 introduction, SVAEs have seen few applications because their expressivity leads to optimization challenges. This work proposes three key fixes that enable efficient training of general SVAEs.

SVAE inference requires iterative optimization [62, 20] of variational parameters for latent variables associated with every observation. Johnson et al. [29] backpropagate gradients through this multi-stage optimization, incurring prohibitive memory cost. We resolve this issue via an implicit differentiation scheme that shows empirical robustness even when inference has not fully converged. Prior work [29] also identifies natural gradients [2, 23] as an important accelerator of optimization convergence, but apply natural gradients in a manner that requires dropping parts of the SVAE loss, yielding biased learning updates. We instead derive unbiased natural gradient updates that are easily and efficiently implemented for any SVAE model via automatic differentiation.

Basic VAEs require carefully tuned continuous relaxations [27, 44] for discrete latent variables, but SVAEs can utilize them seamlessly. We incorporate adaptive variational inference algorithms [24, 25] to robustly avoid local optima when learning SVAEs with discrete structure, enabling data clustering. SVAE inference easily accommodates missing data, leading to accurate and multimodal imputations. We further improve training speed by generalizing prior work on parallel Kalman smoothers [56].

37th Conference on Neural Information Processing Systems (NeurIPS 2023).

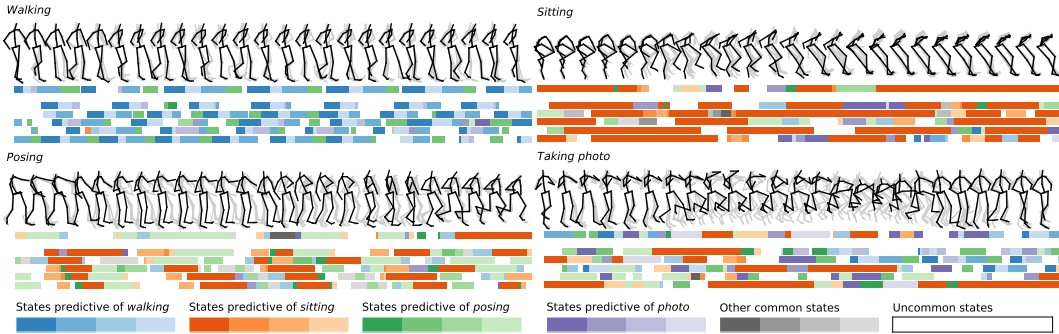

Figure 1: The SVAE-SLDS segments each sequence of human motion, which we display as a sequence of discrete colors. *Discrete variables are interpretable:* Below each segmentation, we show five segmentations of *other* subjects performing the same action, noting similarity across semantically similar series. *Discrete variables are compact representations:* Drawing multiple samples from the generative model conditioned on ground-truth segmentations yields the stick figures in grey, which track closely with the observed data.

We begin in Sec. 2 and 3 by linking variational inference in graphical models and VAEs. Our optimization innovations (implicit differentiation in Sec. 4, unbiased natural gradients in Sec. 5, variational inference advances in Sec. 6) then enable SVAE models to be efficiently trained to their full potential. Although SVAEs may incorporate any latent graphical structure, we focus on temporal data. In Sec. 8, we are the first to compare SVAE performance to state-of-the-art recurrent neural network- and transformer-based architectures on time series benchmarks [21], and the first to demonstrate that SVAEs provide a principled method for multimodal interpolation of missing data.

## 2 Background: Graphical Models and Variational Inference

We learn generative models that produce complex data $x$ via lower-dimensional latent variables $z$. The distribution $p(z|\theta)$ is defined by a graphical model (as in Fig. 2) with parameters $\theta$, and $z$ is processed by a (deep) neural network with weights $\gamma$ to compute the data likelihood $p_\gamma(x|z)$.

Exact evaluation or simulation of the posterior $p_\gamma(z, \theta|x)$ is intractable due to the neural network likelihood. *Variational inference* (VI [62]) defines a family of approximate posteriors, and finds the distribution that best matches the true posterior by optimizing the *evidence lower bound* (ELBO):

$$\mathcal{L}[q(\theta;\eta)q(z;\omega), \gamma] = \mathbb{E}_{q(\theta;\eta)q(z;\omega)}\left[\log \frac{p(\theta)p(z|\theta)p_\gamma(x|z)}{q(\theta;\eta)q(z;\omega)}\right] \leq \log p_\gamma(x). \quad (1)$$

Here, $q(\theta;\eta)q(z;\omega) \approx p_\gamma(z, \theta|x)$ are known as *variational factors*. We parameterize these distributions via arbitrary exponential families with *natural parameters* $\eta, \omega$. This implies that

$$q(z;\omega) = \exp\{\langle \omega, t(z) \rangle - \log Z(\omega)\}, \qquad Z(\omega) = \int_z \exp\{\langle \omega, t(z) \rangle\} \, dz. \quad (2)$$

An exponential family is log-linear in its sufficient statistics $t(z)$, where the normalizing constant $Z(\omega)$ ensures it is a proper distribution (see App. C.1 for properties of exponential families). For models where $p_\gamma(x|z)$ has a restricted conjugate form (rather than a deep neural network), we can maximize Eq. (1) by alternating optimization of $\eta, \omega$; these coordinate ascent updates have a closed form [62]. *Stochastic VI* [23] improves scalability (for models with exponential-family likelihoods) by sampling batches of data $x$, fitting a locally-optimal $q(z;\omega)$ to the latent variables in that batch, and updating $q(\theta;\eta)$ by the resulting (natural) gradient.

**Amortized VI.** Because it is costly to optimize Eq. (1) with respect to $\omega$ for each batch of data, VAEs employ *amortized VI* [32, 47, 53] to approximate the parameters of the optimal $q(z;\omega)$ via a neural network *encoding* of $x$. The inference network weights $\phi$ for this approximate posterior $q_\phi(z|x)$ are jointly trained with the generative model. A potentially substantial *amortization gap* exists [14, 36]: the inference network does not globally optimize the ELBO of Eq. (1) for all $x$.

**Structured VAEs.** For a fixed $q(\theta)$, the true optimizer (across all probability distributions) of Eq. (1) is given by $q(z) \propto p_0(z;\eta)p_\gamma(x|z)$, where $p_0(z;\eta) \propto \exp\{\mathbb{E}_{q(\theta;\eta)}[\log p(z|\theta)]\}$ is an expected prior on $z$ (see App. C.2 for derivations). This simple product of expected prior and likelihood cannot be normalized because of the neural network parameterization of $p_\gamma(x|z)$. Rather than approximating

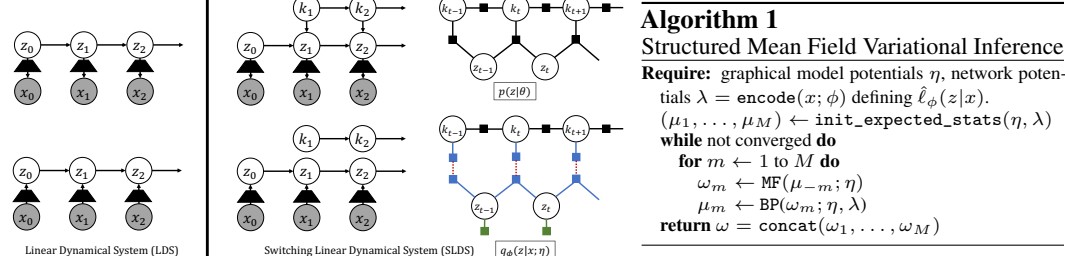

Figure 2: *Left:* Generative (above) and inference (below) graphical models for SVAE-LDS and SVAE-SLDS. For the SLDS, we show the prior and posterior as factor graphs [38]. $q_\phi(z|x)$ combines potentials from the inference network with the true prior. Structured variational inference separates continuous from discrete latent variables for tractability, and mean field messages are propagated across residual edges between disentangled factors. *Right:* General form of iterative mean-field (MF) and belief-propagation (BP) updates for SVAE inference in a model where $q(z)$ factorizes into $M$ groups of latent variables. For the SVAE-SLDS, $M = 2$.

this whole posterior as in amortized VI, the SVAE [29] only approximates the likelihood function, and explicitly multiplies it by the known expected prior to obtain an approximate posterior.

In more detail, the SVAE approximates the likelihood $\ell(z) = p_\gamma(x|z)$ with a function parameterized by a neural network $\hat{\ell}_\phi(z|x)$. The optimal posterior *given this approximation* is then equal to

$$q_\phi(z|x;\eta) = \underset{q(z)}{\arg\max}\, \hat{\mathcal{L}}[q(\theta;\eta)q(z), \phi] = \underset{q(z)}{\arg\max}\, \mathbb{E}_{q(\theta;\eta)q(z)}\left[\log\frac{p(\theta)p(z|\theta)\hat{\ell}_\phi(z|x)}{q(\theta;\eta)q(z)}\right]. \quad (3)$$

The posterior that optimizes the *surrogate loss* $\hat{\mathcal{L}}[\cdot, \phi]$ of Eq. (3) is $q_\phi(z|x;\eta) \propto p_0(z;\eta)\hat{\ell}_\phi(z|x)$. If $\hat{\ell}_\phi(z|x)$ is chosen to be conjugate to $p(z|\theta)$, this multiplication and normalization is easy for many exponential families. Note that $\hat{\ell}_\phi(z|x)$ does not need to be normalized, as any multiplicative factors would disappear when the posterior is normalized. The overall ELBO is then $\mathcal{L}[q(\theta;\eta)q_\phi(z|x;\eta), \gamma]$.

If the encoder's approximate likelihood $\hat{\ell}_\phi$ is (up to normalization) close to the true likelihood $\ell$, the surrogate loss $\hat{\mathcal{L}}$ will closely approximate the true loss $\mathcal{L}$, and there will be little amortization gap. SVAE inference has the advantage that $q_\phi(z|x;\eta)$ depends on the learned posterior $q(\theta;\eta)$ of graphical model parameters, so as learning improves the graphical generative model, the coupled inference model remains closely aligned with the generative process. The ladder VAE [57] and related hierarchical VAEs [59, 12] also incorporate generative parameters in amortized variational inference, but impose a restricted generative hierarchy that is less flexible and interpretable than the SVAE.

**Example 1: Standard Normal.** For a basic VAE, $\theta$ is fixed and $z \sim \mathcal{N}(0, I)$. The SVAE inference network outputs a Gaussian $\hat{\ell}_\phi(z \mid \mu = \mu(x;\phi), \tau = \tau(x;\phi)) \propto \mathcal{N}(z;\mu, \text{diag}(\tau^{-1}))$ with mean $\mu$ and inverse-variance (precision) $\tau$. The product of this Gaussian with the standard normal prior has a simple form: $q_\phi(z|x) = \mathcal{N}(z; \frac{\tau}{\tau+1}\mu, \text{diag}((\tau+1)^{-1}))$. This reparameterization of the standard VAE posterior imposes the (useful) constraint that posterior variances must be smaller than prior variances.

**Example 2: Linear Dynamical System (LDS).** A LDS model for temporal data assumes the latent *state* variables evolve linearly with Gaussian noise: $z_t \sim \mathcal{N}(A_t z_{t-1} + b_t, Q_t)$, $z_1 \sim \mathcal{N}(\mu_1, \Sigma_1)$. In this case, we expand the exponential family distribution $q(z;\omega)$ as a sum across time steps:

$$q(z;\omega) = q(z_1)\prod_{t=2}^{T} q(z_t|z_{t-1}) = \exp\left\{\langle\omega_1, t(z_1)\rangle + \sum_{t=2}^{T}\langle\omega_t, t(z_{t-1}, z_t)\rangle - \log Z(\omega)\right\}, \quad (4)$$

where $\omega = \texttt{concat}(\omega_t)$ and the prior $p(z|\theta)$ belongs to this family. The prior induces temporal dependence between the $z_t$, but we assume the likelihood factorizes as $p_\gamma(x|z) = \prod_t p_\gamma(x_t|z_t)$.

For models like the LDS, approximating the likelihood function with conjugate, time-independent Gaussian distributions is a much simpler task than approximating the temporally-coupled posterior. In addition, as the generative parameters $A_t, b_t, Q_t$ of $p(z|\theta)$ are learned through optimization of $q(\theta)$, the inference routine shares those parameters, improving accuracy. These advantages were not present in the standard normal VAE, where the prior on $z$ lacks structure and is not learned. Inference, normalization, and sampling of $q_\phi(z|x;\eta)$ in the LDS model is feasible via a Kalman smoothing algorithm [4] that efficiently (with cost linear in $T$) aggregates information across time steps.

# 3 Structured Variational Inference

For complex graphical models, the distributions $p(z|\theta)$ and $q(z)$ typically factorize across subsets of the latent variables $z$, as illustrated in Fig. 2. We thus generalize Eq. (4) by partitioning $z$ into local variable groups, and representing the dependencies between them via a set of factors $\mathcal{F}$:

$$q(z;\omega) = \exp\Big\{ \sum_{f \in \mathcal{F}} \langle \omega_f, t(z_f) \rangle - \log Z(\omega) \Big\}. \tag{5}$$

For certain factor graphs, we can efficiently compute marginals and draw samples via the *belief propagation* (BP) algorithm [48, 38]. However, exact inference is intractable for many important graphical models, making it impossible to compute marginals or normalize the $q_\phi(z|x;\eta)$ defined in Sec. 2. SVAE training addresses this challenge via *structured* variational inference [20, 63, 62], which optimizes the surrogate loss across a restricted family of tractable distributions. We connect structured VI to SVAEs in this section, and provide detailed proofs in App. C.2.

## 3.1 Background: Block Coordinate Ascent for Mean Field Variational Inference

Let $\{z_m\}_{m=1}^M$ be a partition of the variables in the graphical model, chosen so that inference within each $z_m$ is tractable. We infer factorized (approximate) marginals $q_\phi(z_m|x;\eta)$ for each mean field cluster by maximizing $\hat{\mathcal{L}}[q(\theta;\eta) \prod_m q(z_m), \phi]$. The optimal $q_\phi(z_m|x;\eta)$ inherit the structure of the joint optimizer $q_\phi(z|x;\eta)$, replacing any factors which cross cluster boundaries with factorized approximations (see Fig. 2). The optimal parameters for these disentangled factors are a linear function of the expected statistics of clusters connected to $m$ via residual edges. These expectations in turn depend on their clusters' parameters, defining a stationary condition for the optimal $\omega$:

$$\omega_m = \texttt{MF}(\mu_{-m};\eta), \qquad \mu_m = \texttt{BP}(\omega_m;\eta,\phi,x). \tag{6}$$

Here, BP is a belief propagation algorithm which computes expected statistics $\mu_m$ for cluster $m$, and the linear *mean field* function MF updates parameters of cluster $m$ given the expectations of *other* clusters $\mu_{-m}$ along residual edges. We solve this optimization problem via the *block updating* coordinate ascent in Alg. 1, which is guaranteed to converge to a local optimum of $\hat{\mathcal{L}}[q(\theta) \prod_m q(z_m)]$.

## 3.2 Reparameterization and Discrete Latent Variables

While optimizing $q_\phi(z_m|x;\eta)$ at inference time requires some computational overhead, it allows us to bypass the typical obstacles to training VAEs with discrete latent variables. To learn the parameters $\phi$ of the inference network, conventional VAE training backpropagates through samples of latent variables via a smooth reparameterization [32], which is impossible for discrete variables. Many alternatives either produce biased gradients [6] or extremely high-variance gradient estimates [50, 28]. Continuous relaxations of discrete variables [44, 27, 7] produce biased approximations of the true discrete ELBO, and are sensitive to annealing schedules for temperature hyperparameters.

SVAE training only requires reparameterized samples of those latent variables which are direct inputs to the generative network $p_\gamma(x|z)$. By restricting these inputs to continuous variables, and using other discrete latent variables to capture their dependencies, discrete variables are marginalized via structured VI *without* any need for biased relaxations. With a slight abuse of notation, we will denote continuous variables in $z$ by $z_m$, and discrete variables by $k_m$.

**Example 3: Gaussian Mixture.** Consider a generalized VAE where the state is sampled from a mixture model: $k \sim \text{Cat}(\pi)$, $z \sim \mathcal{N}(\mu_k, \Sigma_k)$. The likelihood $p_\gamma(x|z)$ directly conditions on only the continuous latent variable $z$. Variational inference produces disentangled factors $q_\phi(z|x;\eta)q_\phi(k|x;\eta)$, and we evaluate likelihoods by decoding samples from $q_\phi(z|x;\eta)$, without sampling $q_\phi(k|x)$.

**Example 4: Switching Linear Dynamical System (SLDS).** Consider a set of discrete states which evolve according to a Markov chain $k_1 \sim \text{Cat}(\pi_0)$, $k_t \sim \text{Cat}(\pi_{k_{t-1}})$, and a continuous state evolving according to switching linear dynamics: $z_0 \sim \mathcal{N}(\mu_0, \Sigma_0)$, $z_t \sim \mathcal{N}(A_{k_t} z_{t-1} + b_{k_t}, Q_{k_t})$. The transition matrix, offset, and noise at step $t$ depends on $k_t$. Exact inference in SLDS is intractable [39], but structured VI [20] learns a partially factorized posterior $q_\phi(z|x;\eta)q_\phi(k|x;\eta)$ that exactly captures dependencies *within* the continuous and discrete Markov chains.

BP for SLDS uses variational extensions [5, 4] of the Kalman smoother to compute means and variances of continuous states, and forward-backward message-passing to compute marginals of discrete states (see App. D). Let $k_{tj} = 1$ if the SLDS is in discrete state $j$ at time $t$, $k_{tj} = 0$ otherwise,

| | Time of gradient step (ms) | | | |
|---|---|---|---|---|
| Method | $B = 1$ | $B = 32$ | $B = 64$ | $B = 128$ |
| Implicit + Parallel | 603 | 922 | 1290 | 2060 |
| Unrolled + Parallel | 659 | 1080 | n/a | n/a |
| Implicit + Sequential | 2560 | 3160 | 3290 | 3530 |
| Unrolled + Sequential | 2660 | 3290 | 3980 | n/a |

Table 1: Time of ELBO backpropagation in an SVAE-SLDS with $K = 50$ discrete states, dimension $D = 16$, and $T = 250$ time steps. For varying batch sizes $B$, we compare *capped implicit* gradients to unrolled gradients for $L = 10$ block updates of two inference algorithms: standard sequential BP, and our parallel extension. For large batch sizes, unrolled gradients crashed because it attempted to allocate more than 48GB of GPU memory.

and $\bar{\theta}_j = \mathbb{E}_{q(\theta;\eta)}[\theta_j]$ be the expected (natural) parameters of the LDS for discrete state $j$. Structured VI updates the natural parameters of discrete states $\omega_{k_{tj}}$, and continuous states $\omega_{z_t,z_{t+1}}$, as follows:

$$\omega_{z_t,z_{t+1}} = \sum_j \mathbb{E}_q[k_{tj}]\bar{\theta}_j, \qquad \omega_{k_{tj}} = \langle \bar{\theta}_j, \mathbb{E}_q[t(z_{t-1}, z_t)]\rangle. \tag{7}$$

## 4  Stable and Memory-Efficient Learning via Implicit Gradients

When $q_\phi(z|x)$ is computed via closed-form inference, gradients of the SVAE ELBO may be obtained via automatic differentiation. This requires backpropagating through the encoder and decoder networks, as well as through reparameterized sampling $z \sim q_\phi(z|x;\eta)$ from the variational posterior.

For more complex models where structured VI approximations are required, gradients of the loss become difficult to compute because we must backpropagate through Alg. 1. For the SLDS this *unrolled* gradient computation must backpropagate through repeated application of the Kalman smoother and discrete BP, which often has prohibitive memory cost (see Table 1).

We instead apply the *implicit function theorem* (IFT [34]) to compute implicit gradients $\frac{\partial\omega}{\partial\eta}, \frac{\partial\omega}{\partial\phi}$ without storing intermediate states. We focus on gradients with respect to $\eta$ for compactness, but gradients with respect to $\phi$ are computed similarly. Let $\omega^{(1)}, \ldots, \omega^{(L)}$ be the sequence of $\omega$ values produced during the "forward" pass of block coordinate ascent, where $\omega^{(L)}$ are the optimized structured VI parameters. The IFT expresses gradients via the solution of a set of linear equations:

$$\frac{\partial\omega^{(L)}}{\partial\eta} = \left(\frac{\partial g(\omega;\eta,\phi,x)}{\partial\omega}\right)^{-1}\frac{\partial g(\omega;\eta,\phi,x)}{\partial\eta}, \qquad g(\omega) = \omega - \mathtt{MF}(\mathtt{BP}(\omega;\eta,\phi,x);\eta). \tag{8}$$

Here we apply the BP and MF updates in *parallel* for all variable blocks $m$, rather than sequentially as in Eq. (6). At a VI fixed point, these parallel updates leave parameters unchanged and $g(\omega) = 0$.

For an SLDS with latent dimension $D$ and $K$ discrete states, $\omega$ has $O(K + D^2)$ parameters at each time step. Over $T$ time steps, $\frac{\partial g}{\partial\omega}$ is thus a matrix with $O(T(D^2 + K))$ rows/columns and $O(T^2D^2K)$ non-zero elements. For even moderate-sized models, this is infeasible to explicitly construct or solve.

We numerically solve Eq. (8) via a Richardson iteration [54, 64] that repeatedly evaluates matrix-vector products $(I - A)v'$ to solve $A^{-1}v$. Such numerical methods have been previously used for other tasks, like hyperparameter optimization [43] and meta-learning [49], but not for the training of SVAEs. The resulting algorithm resembles unrolled gradient estimation, but we repeatedly backpropagate through updates at the *endpoint* of optimization instead of along the optimization trajectory.

$$\textit{Richardson:} \qquad \frac{\partial\omega^{(L)}}{\partial\eta} \approx -\sum_{j=0}^{J}\left(I - \frac{\partial g(\omega^{(L)};\eta,\phi,x)}{\partial\omega}\right)^j\frac{\partial g(\omega^{(L)};\eta,\phi,x)}{\partial\eta}. \tag{9}$$

$$\textit{Unrolled:} \qquad \frac{\partial\omega^{(L)}}{\partial\eta} \approx -\sum_{\ell=0}^{L}\left[\prod_{i=\ell}^{L}\left(I - \frac{\partial g(\omega^{(i)};\eta,\phi,x)}{\partial\omega}\right)\right]\frac{\partial g(\omega^{(\ell)};\eta,\phi,x)}{\partial\eta}. \tag{10}$$

Lorraine et al. [43] tune the number of Richardson steps $J$ to balance speed and accuracy. However, there is another reason to limit the number of iterations: when the forward pass is not iterated until convergence, $\omega^{(L)}$ is not a stationary point of $g(\omega)$ and therefore Eq. (9) is not guaranteed to converge as $J \to \infty$. For batch learning, waiting for *all* VI routines to converge to a (local) optimum might be prohibitively slow, so we might halt VI before $\omega^{(L)}$ converges to numerical precision.

Seeking robustness even when the forward pass has not converged, we propose a *capped implicit* gradient estimator that runs one Richardson iteration for every step of forward optimization, so that

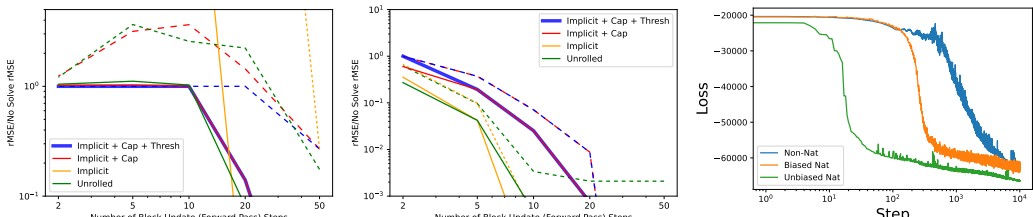

Figure 3: We compare implicit gradient estimators' stability (left, middle), and gradient conditioning methods' loss trajectory (right), on human motion capture data (Sec. 8). *Stability:* Gradient estimate rMSE relative to the *No-Solve* estimator (smaller is better) for various numbers of VI block updates $L$, and SVAE-SLDS models taken from the start of training (left) and after 20 epochs (middle). Solid lines show median rMSE ratio across a batch of 128 data points, and dashed lines show $90^{\text{th}}$ percentiles. *Conditioning:* Convergence of SVAE-LDS negative-ELBO loss versus number of optimization steps (log-scale) for conventional (non-natural) gradients, biased natural gradients [29], and unbiased natural gradients computed via automatic differentiation (Sec. 5).

$J = L$. In this regime, implicit gradient computation has a one-to-one correspondence to unrolled gradient computation, while requiring a small fraction of the memory. This can be thought of as a form of gradient regularization: if we take very few steps in the forward pass, we should have low confidence in the optimality of our end-point and compute fewer terms of the Neumann series (9).

**Experiments.** In Fig. 3 (left, middle) we compute the accuracy of different approximate gradient estimators for training a SVAE-SLDS as in Fig. 2. To our knowledge, we are the first to investigate the quality of implicit gradients evaluated away from an optimum, and we compare our *capped implicit* proposal to other gradient estimators. Ground truth gradients are computed as the implicit gradient at the optimum, and we compare the root-mean-squared-error (rMSE) of various gradient estimators to that of the naïve *No-Solve* solution, which replaces the inverted matrix in Eq. (8) with Identity.

We consider two models: one with randomly initialized parameters, and one that has been trained for 20 epochs. The newly-initialized model requires more forward steps for the block updating routine to converge. We compare the memory-intensive unrolled estimator (*Unrolled*) to three versions of the implicit gradient estimator. First, an uncapped version (*Implicit*) always performs $J = 50$ Richardson iterations regardless of the number of forward iterations, thus incurring high computation time. Note that evaluating implicit gradient far from an optimum can produce high error; in the newly-initialized model, many of these iterations diverge to infinity when fewer than 20 forward steps are taken. Second, we consider a capped implicit estimator (*Implicit+Cap*) which sets $J = L$ to match the number of forward steps. Finally, we consider a capped implicit estimator which also includes a threshold (*Implicit+Cap+Thresh*): if the forward pass has not converged in the specified number of steps, the thresholded estimator simply returns the *No-Solve* solution. This gradient is stable in all regimes while retaining desirable asymptotic properties [64]. Our remaining experiments therefore use this method for computing gradients for SVAE training.

**Prior work.** Johnson et al. [29] consider implicit differentiation, but only very narrowly. They derive implicit gradients by hand in cases (like the LDS) where exact inference is tractable, so the linear solve in Eq. (8) cancels with other terms, and gradients may be evaluated via standard automatic differentiation. For models requiring structured VI (like the SLDS), [29] instead computes *unrolled* gradients for inference network weights $\phi$, suffering high memory overhead. They compute neither unrolled nor implicit gradients with respect to generative model parameters $\eta$; in practice they set the gradient of the inner optimization to 0, yielding a biased training signal. Our innovations instead enable memory-efficient and unbiased gradient estimates for all parameters, for all graphical models.

## 5 Rapid Learning via Unbiased Natural Gradients

SVAE training must optimize the parameters of probability distributions. Gradient descent implicitly uses Euclidean distance as its notion of distance between parameter vectors, which is often a poor indicator of the divergence between two distributions. The natural gradient [2] resolves this issue by rescaling the gradient by the Fisher information matrix $F_\eta$ of $q(\theta; \eta)$, given by:

$$F_\eta = \mathbb{E}_{q(\theta;\eta)}\big[\big(\nabla_\eta q(\theta;\eta)\big) \cdot \big(\nabla_\eta q(\theta;\eta)\big)^T\big]. \tag{11}$$

Johnson et al. [29] demonstrate the advantages of natural gradients for the SVAE, drawing parallels to the natural gradients of stochastic VI (SVI [23]). SVI extends the variational EM algorithm to mini-batch learning: similar to the SVAE, it fits $q(z)$ in an inner optimization loop and learns $q(\theta;\eta)$ in an outer loop by natural gradient descent. The key difference between SVI and the SVAE is that SVI's inner optimization is done with respect to the true loss function $\mathcal{L}$, whereas the SVAE uses a surrogate $\hat{\mathcal{L}}$. SVI can only do this inner optimization by restricting all distributions to be conjugate exponential family members, giving up the flexibility provided by neural networks in the SVAE.

Let $\mu_\eta$ be the expected sufficient statistics of $q(\theta;\eta)$. Exponential family theory tells us that $\frac{\partial \mu}{\partial \eta} = F_\eta$ [30, 45], allowing Johnson et al. [29] to derive the natural gradients of the SVAE loss:

$$\frac{\partial \mathcal{L}}{\partial \eta} F_\eta^{-1} = \frac{\partial \mathcal{L}}{\partial \mu}\frac{\partial \mu}{\partial \eta} F_\eta^{-1} = \frac{\partial \mathcal{L}}{\partial \mu}, \qquad \frac{\partial \mathcal{L}}{\partial \mu} = \overbrace{\eta_0 + \mathbb{E}_{q_\phi(z|x;\eta)}[t(z)] - \eta}^{\text{SVI update}} + \overbrace{\frac{\partial \mathcal{L}}{\partial \omega} \cdot \frac{\partial \omega}{\partial \eta}}^{\text{correction term}} . \quad (12)$$

This gradient differs from the SVI gradient by the final term: because SVI's inner loop optimizes $\omega$ with respect to the true loss $\mathcal{L}$, $\frac{\partial \mathcal{L}}{\partial \omega} = 0$ for conjugate models. Johnson et al. [29] train their SVAE by dropping the correction term and optimizing via the SVI update equation, yielding biased gradients.

There are two challenges to computing unbiased gradients in the SVAE. First, in the structured mean field case $\frac{\partial \omega}{\partial \eta}$ involves computing an implicit or unrolled gradient, as addressed by our numerical methods in Sec. 4. Second, including the correction term in the gradient costs us a desirable property of the SVI natural gradient: for step size less than 1, any constraints on the distribution's natural parameters are guaranteed to be preserved, such as positivity or positive-definiteness.

We resolve this issue by reparameterizing $\eta$ into an unconstrained space, and computing natural gradients with respect to those new parameters. Letting $\tilde{\eta}$ be an unconstrained reparameterization of $\eta$, such as $\eta = \texttt{Softplus}\{\tilde{\eta}\} = \log(1 + e^{\tilde{\eta}})$ for a positive precision parameter, we have:

$$\frac{\partial \mathcal{L}}{\partial \tilde{\eta}} F_{\tilde{\eta}}^{-1} = \frac{\partial \mathcal{L}}{\partial \mu} \cdot \frac{\partial \eta}{\partial \tilde{\eta}}^{-T} = \left(\frac{\partial \tilde{\eta}}{\partial \eta} \cdot \nabla_\mu \mathcal{L}\right)^T . \quad (13)$$

See App. F for proof. This differs from the non-natural gradient in two ways. First, the Jacobian of the $\eta \to \mu$ map is dropped, as before. Unlike Johnson et al. [29], we do not hand-derive the solution; we employ a *straight-through gradient estimator* [6] to replace this Jacobian with the identity. Then, the Jacobian of the $\tilde{\eta} \to \eta$ map is replaced by its inverse-transpose. This new gradient can be computed without any matrix arithmetic by noting that the inverse of a Jacobian is the Jacobian of the inverse function. Thus Eq. (13) can be computed by replacing the reverse-mode backpropagation through the $\tilde{\eta} \to \eta$ map with a forward-mode differentiation through the inverse $\eta \to \tilde{\eta}$ map.

In Fig. 3 (right) we show the performance benefits of our novel unbiased natural gradients with stochastic gradient descent, compared to regular gradients with an Adam optimizer [31], and stochastic gradient descent via biased natural gradients [29] that drop the correction term. Results are shown for an SVAE-LDS model whose pre-trained encoder and decoder are fixed.

## 6  Adapting Graphical Model Innovations

Efficient implementations of BP inference, parameter initializations that avoid poor local optima, and principled handling of missing data are well-established advantages of the graphical model framework. We incorporate all of these to make SVAE training more efficient and robust.

**Parallel inference.**  The BP algorithm processes temporal data sequentially, making it poorly suited for large-scale learning of SVAEs on modern GPUs. Särkkä & García-Fernández [56] developed a method to parallelize the usually-sequential Kalman smoother algorithm across time for jointly Gaussian data. Their algorithm is not directly applicable to our VI setting where we take expectations over $q(\theta)$ instead of having fixed parameters $\hat{\theta}$, but we derive an analogous parallelization of variational BP in App. D.2. We demonstrate large speeds gains from this adaptation in Table 1.

**Initialization.**  Poor initialization of discrete clusters can cause SLDS training to collapse to a single discrete state. This problem becomes worse when the graphical model is trained on the output of a neural network encoder, which when untrained produces outputs which do not capture meaningful statistics of the high-dimensional data. We therefore propose a three-stage training routine: a basic VAE is trained to initialize $p_\gamma(x|z)$, and then the output of the corresponding inference network is used for variational learning of graphical model parameters [25]. Once the deep network and graphical model are sensibly initialized, we refine them via joint optimization while avoiding collapse. For details of this initialization scheme, see App. A.5.

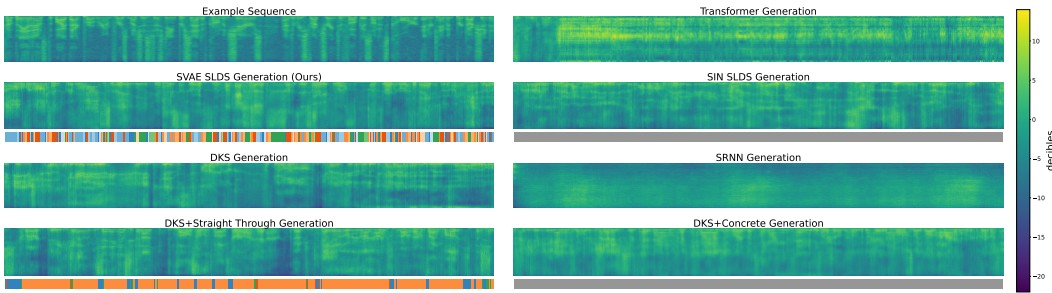

Figure 4: Unconstrained generation of 513-dim. speech spectrogram data over $T = 500$ time-steps (horizontal; models are trained on data with $T = 50$). An example sequence of real speech data is shown. For models which use discrete latent variables, the sequence of discrete states is shown as a changing colorbar beneath the generation, with a solid colorbar meaning a constant discrete state for the entire sequence.

**Missing data.** The structure provided by the SVAE graphical model allows us to solve marginal and conditional inference queries not seen at training time. In particular, we explore the ability of a trained SVAE to impute data that is missing for an extended interval of time. By simply setting $\hat{\ell}_\phi(z_t|x_t; \eta)$ to be uniform at a particular timestep $t$, our posterior estimate of $z_t$ is only guided by the prior, which aggregates information across time to produce a smooth estimate of the posterior on $z_t$.

While discriminative methods may be explicitly trained to impute time series, we use imputation performance as a measure of generative model quality, so do not compare to these approaches. Unlike discriminative methods, SVAE imputation does *not* require training data with aligned missing-ness.

## 7 Related Work

**Dynamical VAEs.** Girin et al. [21] provide a comprehensive survey of dynamical VAEs (DVAEs) for time series data, which use recurrent neural networks to model temporal dependencies. The *Stochastic Recurrent Neural Network* (SRNN [17]), which has similar structure to the *Variational Recurrent Neural Network* (VRNN [13]), is the highest-performing model in their survey; it models data via one-step-ahead prediction, producing probabilities $p(x_t|z, x_{t-1})$. This model therefore reconstructs $x$ using more information than is encoded in $z$ by skipping over the latent state and directly connecting ground truth to reconstruction, reducing the problem of sequence generation to a series of very-local one-step predictions. On the other hand, the *Deep Kalman Smoother* (DKS [35]) extension of the Deep Kalman Filter [37] is the best-performing model which generates observations independently across time, given only information stored in the latent encoding $z$.

RNNs lack principled options for handling missing data. Heuristics such as constructing a dummy neural network input of all-zeros for unobserved time steps, or interpolating with exponentially decayed observations [11], effectively require training to learn these imputation heuristics. RNNs must thus be trained on missing-ness that is similar to test missing-ness, unlike the SVAE.

Transformers [61] have achieved state-of-the-art generative performance on sequential language modeling. However, Zeng et al. [66] argue that their permutation-invariance results in weak performance for time-series data where each observation carries low semantic meaning. Unlike text, many time series models are characterized by their temporal dynamics rather than a collection of partially-permutable tokens. Lin et al. [42] propose a dynamical VAE with encoder $q(z_t|x_{1:T})$, decoder $p(x_{t+1}|x_{1:t}, z_{1:t+1})$, and latent dynamics $p(z_{t+1}|x_{1:t}, z_{1:t})$ parameterized by transformers.

**Structured VAEs.** We, as in Johnson et al. [29], only consider SVAEs where the inference network output factorizes across (temporal) latent variables. Orthogonal to our contributions, Yu et al. [65] investigate the advantages of taking the SVAE beyond this restriction, and building models where the recognition network outputs proxy-likelihood functions on *groups* of latent variables.

In recent independent work, Zhao & Linderman [67] also revisit the capabilities of the SVAE. However, their work differs from ours in a few key respects. First, because their experiments are restricted to the LDS graphical model (which requires no mean field factorization nor block updating), they do not need implicit differentiation, and do not explore the capacity of the SVAE to include discrete latent variables. Second, because they optimize point-estimates $\hat{\theta}$ of parameters instead of variational factors $q(\theta)$, they do not make use of natural gradients. In this point-estimate formulation,

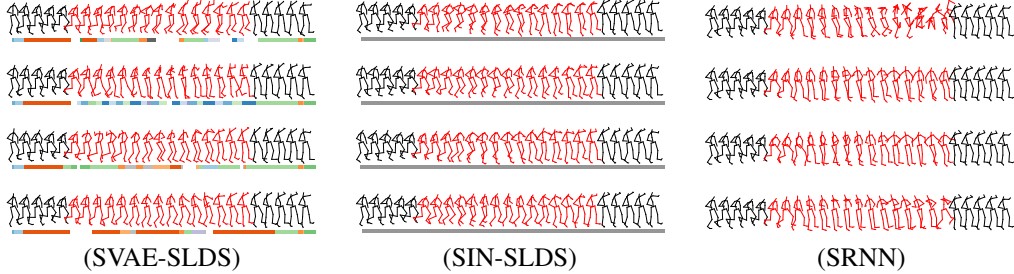

|       (SVAE-SLDS)       |       (SIN-SLDS)       |       (SRNN)       |

Figure 5: Interpolations of human motion capture data. Red figures (covering 150 time steps) are generated by each model to interpolate between the black figures four times. We see that our SVAE-SLDS provides varied and plausible imputations with corresponding segmentations (colors shared with Fig. 1). During training the SIN-SLDS [41] collapses to only use a single discrete state, and thus cannot produce diverse imputations. The SRNN [17] produces varied sequences, but autoregressive generation is sometimes unstable and unrealistic, and its inability to account for future observations prevents smooth interpolation with the observed sequence end.

they apply the parallel Kalman smoother [56] off-the-shelf, whereas we derive a novel extension for our variational setting. Finally, their experimental results are confined to toy and synthetic data sets.

The most directly comparable model to the SVAE is the *Stochastic Inference Network* (SIN [41]), which employs a graphical model prior $p(z|\theta)$ but estimates $q(z)$ through traditional amortized inference; a parameterized function that shares no parameters with the graphical model produces variational posteriors. The authors consider discrete latent variable models like the SLDS, but due to the intractability of discrete reparameterization, their inference routine fits the continuous latent variables with a vanilla LDS. Thus training pushes the SIN to reconstruct, and therefore model, the data without the use of switching states. (Experiments in [41] consider only the LDS, not the SLDS.)

The Graphical Generative Adversarial Network [40] integrates graphical models with GANs for structured data. Experiments in [40] solely used image data; we compared to their implementation, but it performed poorly on our time-series data, generating unrecognizable samples with huge FID.

**Vector Quantization.** Vector-Quantized VAEs (VQ-VAEs [60]) are another family of deep generative models that make use of discrete latent representations. Rather than directly outputting an estimated posterior, the encoder of a VQ-VAE outputs a point in an embedding space which is *quantized* to one of $K$ learnable quantization points. The encoder is trained using a straight-through estimator [6]. While discrete SVAE-SLDS states switch among multiple continuous modes, the VQ-VAE representation is purely discrete, limiting information encoded by a latent variable to $\log_2 K$ bits. In order to generate plausible and diverse samples, VQ-VAEs require large values of $K$, structured collections of discrete variables, and/or post-hoc training of autoregressive priors [51, 52, 16, 15].

## 8   Experiments

We compare models via their test likelihoods, the quality of generated data, and the quality of interpolations. We consider joint positions from human motion capture data (MOCAP [10, 26]) and audio spectrograms from recordings of people reading Wall Street Journal headlines (WSJ0 [19]); see Table 2. MOCAP has 84-dimensional data and training sequences of length $T = 250$. WSJ0 has 513-dimensional data and training sequences of length $T = 50$. See App. A.2 for further details.

Generation quality is judged via a modified *Frechét inception distance* (FID [18]) metric. We replace the InceptionV3 network with appropriate classifiers for motion capture and speech data (see App. A.3). SVAE-SLDS-Bias runs the SVAE as presented by Johnson et al. [29], with unrolled gradients, dropped correction term, sequential Kalman smoother, and no pre-training scheme. We match encoder-decoder architectures for all SVAE and SIN models using small networks (about 100,000 total parameters for motion data). The DKS, SRNN, and Transformer DVAE have approximately 300,000, 500,000, and 1.4 million parameters each; see App. A.5 for details.

To demonstrate the SVAE's capacity to handle discrete latent variables in a principled manner, we compare to two DVAE baselines which incorporate discrete variables via biased gradients: the straight-through estimator [6] and the concrete (Gumbel-softmax) distribution [27, 44]. To our knowledge, no one has successfully integrated either method into dynamical VAEs for temporal data. Thus to make comparison possible, we have devised a new model which adds discrete latent variables to the generative process of the DKS. Specifications for this model is provided in App. A.4. We evaluated this model with both gradient estimators, and reported the results in Table 2.

| Method | $\log p(x) \geq$ ($\uparrow$) | Sample FID ($\downarrow$) | Interpolation FIDs ($\downarrow$) | | |
|---|---|---|---|---|---|
| | | | 0.0-0.8 | 0.2-1.0 | 0.2-0.8 |
| **Human Motion Capture (h3.6m)** | | | | | |
| SVAE-SLDS | 2.39 | **12.3 ± 0.2** | **7.9 ± 0.2** | **7.5 ± 0.2** | **2.8 ± 0.02** |
| SVAE-SLDS-Bias [29] | 2.36 | 34.6 ± 0.7 | 28.8 ± 0.2 | 25.8 ± 0.3 | 6.71 ± 0.12 |
| SVAE-LDS | 2.28 | 34.0 ± 0.3 | 19.3 ± 0.2 | 21.9 ± 0.2 | 7.90 ± 0.13 |
| SIN-SLDS [41] | 2.36 | 33.7 ± 0.4 | 12.38 ± 0.12 | 8.97 ± 0.08 | 3.27 ± 0.05 |
| SIN-LDS [41] | 2.33 | 65.2 ± 1.4 | 18.3 ± 0.2 | 15.5 ± 0.2 | 6.24 ± 0.09 |
| Transformer [42] | 2.82 | 421 ± 11 | 234 ± 9 | 228 ± 5 | 113 ± 5 |
| SRNN [17] | **2.94** | 62.7 ± 0.7 | 43.5 ± 0.7 | 24.2 ± 0.6 | 14.2 ± 0.3 |
| DKS [35] | 2.31 | 136 ± 6 | 46.7 ± 1.7 | 33.3 ± 1.1 | 9.0 ± 0.3 |
| DKS+Concrete | 1.70 | 144 ± 3 | 88 ± 3 | 89 ± 2 | 34.0 ± 1.4 |
| DKS+Straight-Through | 2.09 | 22 ± 3 | 22.6 ± 0.3 | 17.15 ± 0.14 | 13.8 ± 0.2 |
| **Audio Spectrogram (WSJ0)** | | | | | |
| SVAE-SLDS | 1.54 | **9.61 ± 0.15** | **7.5 ± 0.2** | 8.14 ± 0.12 | **4.88 ± 0.08** |
| SVAE-SLDS-Bias | 1.45 | 18.6 ± 0.2 | 15.0 ± 0.2 | 15.2 ± 0.2 | 7.6 ± 0.12 |
| SVAE-LDS | 1.56 | 19.1 ± 0.3 | 17.9 ± 0.2 | 16.6 ± 0.3 | 7.2 ± 0.3 |
| SIN-SLDS | 1.53 | 20.0 ± 0.4 | 17.2 ± 0.3 | 14.9 ± 0.3 | 9.5 ± 0.2 |
| SIN-LDS | 1.54 | 17.8 ± 0.2 | 17.21 ± 0.11 | 13.2 ± 0.4 | 10.1 ± 0.2 |
| Transformer | 1.88 | 10.0 ± 0.2 | 12.0 ± 0.3 | **8.2 ± 0.2** | 5.7 ± 0.4 |
| SRNN | **1.94** | 23.6 ± 0.3 | 19.4 ± 0.5 | 17.4 ± 0.3 | 12.7 ± 0.4 |
| DKS | 1.55 | 12.9 ± 0.2 | 10.8 ± 0.2 | 10.8 ± 0.14 | 7.7 ± 0.05 |
| DKS+Concrete | 1.45 | 16.6 ± 0.2 | 12.8 ± 0.2 | 11.3 ± 0.2 | 8.2 ± 0.2 |
| DKS+Straight-Through | 1.48 | 15.51 ± 0.13 | 10.07 ± 0.11 | 9.02 ± 0.18 | 6.29 ± 0.13 |

Table 2: Comparison of model performance on log-likelihood (higher is better), FIDs of unconditionally generated samples (lower is better), and FIDs of interpolations on augmented human motion capture and audio spectrogram data. Each interpolation column corresponds to a masking regime where the shown range of percentiles of the data is masked, e.g. 0.0-0.8 means the first 80% of time steps are masked.

**Interpretability.** In Fig. 1 we show the SVAE-SLDS's learned discrete encoding of several joint tracking sequences. While "sitting" sequences are dominated by a single dynamic mode, walking is governed by a cyclic rotation of states. "Posing" and "Taking photo" contain many discrete modalities which are shared with other actions. The discrete sequences provide easily-readable insight into sequences, while also compactly encoding the high-dimensional data.

**Generation.** In Fig. 4 we show example generated sequences from each model of audio data. Like true speech, the SVAE-SLDS moves between discrete structures over time representing individual sounds. In contrast, the SIN-SLDS [41] and DKS+Concrete baselines collapse to a single discrete modality, blending together continuous dynamics. While the DKS+Straight-Through model does not collapse, it uses discrete states too coarsely to inform the high-frequency dynamics of speech.

**Interpolation.** Amortized VI cannot easily infer $q(z_t)$ at time steps where the observations $x_t$ are missing. Thus, given observations at a subset of times $x_{\text{obs}}$, we can encode to obtain $q(z_{\text{obs}})$ and infer the missing latent variables by drawing from the generative prior $p(z_{\text{missing}}|z_{\text{obs}}, \theta)$. Because baseline models parameterize $p(z|\theta)$ by one-directional neural networks, they can only condition $z_{\text{missing}}$ on $z_{\text{obs}}$ at *previous* time steps, leading to discontinuity at the end of the missing window. For further specifications and for details of the SVAE approach described in Sec. 6, see App. A.6. An alternative approach of in-filling missing data with zeros causes models to reconstruct the zeros; see App. Fig. 7.

In Fig. 5, we see the SVAE-SLDS uses discrete states to sample variable interpolations, while the SRNN's one-step-ahead prediction scheme cannot incorporate future information in imputation, producing discontinuities. We also note that despite achieving the highest test likelihoods, the SRNN produces some degenerate sequences when we iterate next-step prediction, and has inferior FID (see Table 2). The SIN-SLDS collapses to a single discrete state in training, resulting in uniform imputations that lack diversity. Example imputations for all models are provided in App. Fig. 6.

**Transformers for time series.** The permutation-invariance of transformers is visible in its varied performance on these two tasks. A lack of sequential modeling can lead to discontinuities in the data sequence which are naturally present in speech. For MOCAP, joint locations are continuous across time, making transformer-generated samples unrealistic (see Fig. 6 and Table 2).

## 9 Discussion

The SVAE is uniquely situated at the intersection of flexible, high-dimensional modeling and interpretable data clustering, enabling models which both generate data and help us understand it. Our optimization innovations leverage automatic differentiation for broad applicability, and provide the foundation for learning SVAEs with rich, non-temporal graphical structure in other domains.

## Acknowledgements

This research supported in part by NSF RI Award No. IIS-1816365, ONR Award No. N00014-23-1-2712, and the HPI Research Center in Machine Learning and Data Science at UC Irvine.

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

# A   Experimental Protocol

## A.1   Implementation

Code can be found at `https://github.com/hbendekgey/SVAE-Implicit`. All methods were implemented with the JAX library [8].

## A.2   MOCAP Dataset

Our motion capture experiments were conducted using a variant of the H3.6M dataset [26]. The original data consists of high-resolution video data of 11 actors performing 17 different scenarios, captured with 4 calibrated cameras. Following the procedure of [21] our tests use the extracted skeletal tracks, which contain 3D-coordinates for 32 different joints (for 96 observation dimensions), sampled at 25hz. Joint positions are recorded in meters relative to the central pelvis joint. Our only changes from the pre-processing of [21] are (i) we remove dimensions with 0 variance across data sequences, resulting in 84 observation dimensions; (ii) we extract 250-step-long sequences instead of 50-step-long ones, corresponding to an increase from 2 seconds of recording to 10 seconds; (iii) we add Gaussian noise with variance of 1mm to the data to reduce overfitting; and (iv) we employ a different train-valid-test split. In total, our training dataset consisted of 53,443 sequences, our validation set contained 2,752 sequences, and our test set contained 25,893 sequences.

While [21] split across subjects, we found this lead to substantially different training, validation, and testing distributions. We instead split across "sub-acts", or the 2 times each subject did each action. Thus the training set contained 7 actors doing 17 actions once, and the validation and test set combined contained the second occurrence of each act. The validation set contained 3 subjects' second repetition of each action, and the test set contained the other 4. Two corrupted sequences were found in the validation set, *7-phoning-1* and *7-discussion-1*, which were removed.

We model the likelihood $p(x|z)$ of joint coordinates using independent Gaussian distributions. Following the work of [55] we fit a single global variance parameter to each feature dimension (as part of training), rather than fixing the likelihood variances or outputting them from the decoder network. We found this leads to more stable training for most methods.

**FID Scores**   The InceptionV3 [58] network typically used to compute *Frechet Inception Distance* [18] scores was trained on natural images and is therefore inappropriate for evaluating the generative performance of the H3.6M motion capture data. To resolve this, we compute the same score using a different underlying network. We trained a convolutional network to predict which of the 15 different motions was being performed in each sequence (note that in H3.6M, motions are distinct from actions/scenerios, which may include more subtle variations). We used a residual network architecture, where each residual block consists of 2 1-D convolutional layers with batch normalization and a skip connection. We chose 1-D convolutions (across time) as they are more appropriate for our temporal motion capture data. We used the Optuna library [1] to search across model depths, activations and layer sizes to find a network within our design space that performed optimally on a validation dataset (25% of the original training data). Our final network architecture is summarized in table 3. As with the original FID score, we use the output of final average pooling layer to compute the score.

## A.3   WSJ0 Dataset

The WSJ0 dataset [19] contains recordings of people reading news from the Wall Street Journal. We followed all pre-processing of [21], which resulted in sequences of length 50 with 513-dimensional observations at each time step. We modeled the spectrogram data via a Gamma distribution with fixed concentration parameter 2, equivalent to modeling the complex Fourier transform of the data with a mean 0 complex normal distribution. We remove subject 442 from the test data set, as several of the recordings are substantially different from all others in the train and validation sets: instead of being isolated audios, some of these recordings include background noises and beeps which cannot be fit at training time. The encoder takes as input the log of the data and the decoder outputs the log rate of the gamma distribution, as the data spans many orders of magnitude. In total, our training dataset contained 93,215 sequences, our validation set contained 2,752 sequences, and our test set contained 4,709 sequences.

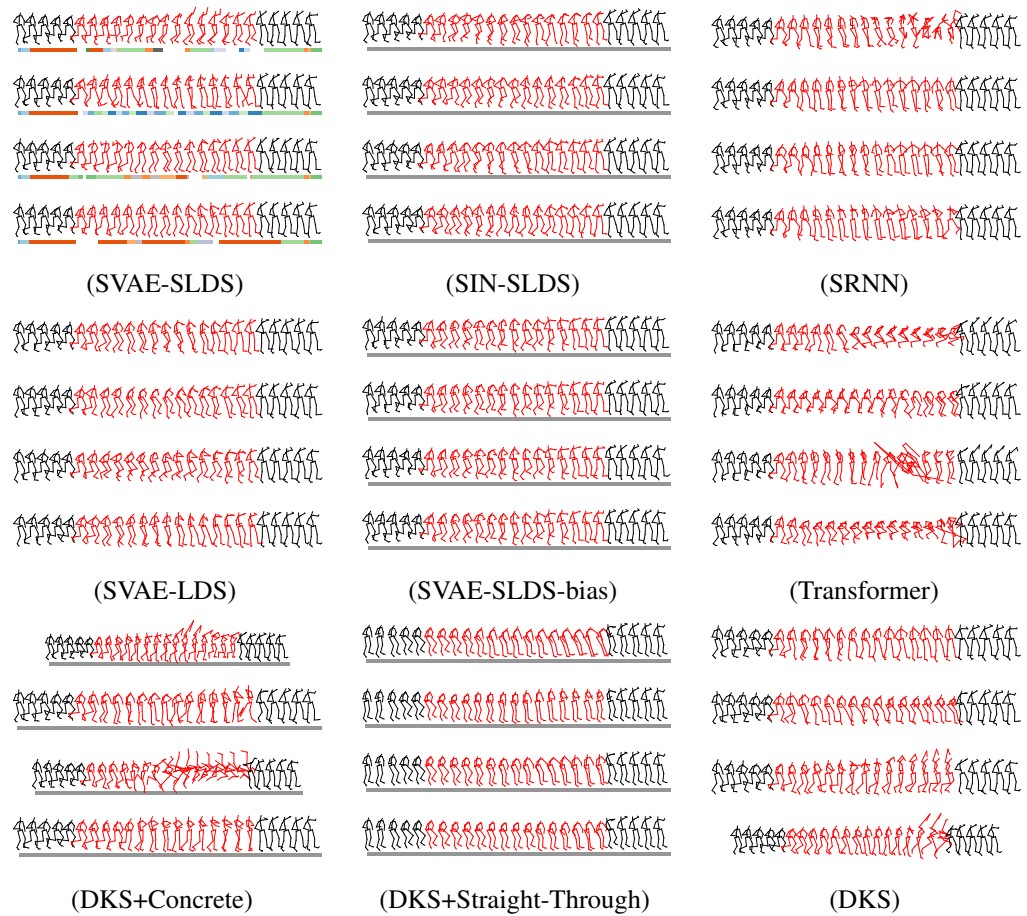

Figure 6: Interpolations of human motion capture data. Red figures (covering 150 time steps) are generated by each model to interpolate between the black figures four times (see App. A.6 for methodology). Segmentation colorbars are shown below models which use discrete latent variables; all non-SVAE-SLDS models collapse to a single discrete state throughout training. Amortized VI models draw missing data conditioned on *previous* data, introducing discontinuities at the end of the missing window.

| Residual Block | Downsampling block | **Mocap Classifier**
Activation: Relu
Conv. window: 3 |
|---|---|---|
| Conv.
Batch Norm
Activation
Conv.
Batch Norm
Activation
Add input | Conv.
Batch Norm
Activation
Conv. (stride 2)
Batch Norm
Activation
Add downsampled input | Residual block 128
Downsampling block 128
Residual block 128
Residual block 128
Global average pool
Dense |

Table 3: Summary of classifier network architecture used to compute FID scores for Mocap data.

**FID Scores**  As with the H3.6M dataset, it is necessary to define a different base network to evaluate generative FID scores for the WSJ0 dataset. As the WSJ0 dataset does not have an appropriate set of labels to use to train a classifier, we chose to instead train a classifier on the CREMA-D dataset [9], which consists of similar speech clips. We preprocessed each audio sequence in CREMA-D identically to the WSJ0 dataset and trained a classifier to predict one of 6 different emotions that was expressed. For architecture, we used a `efficientnet_v2_s` model from the `torchvision` package [46] on the data in log space.

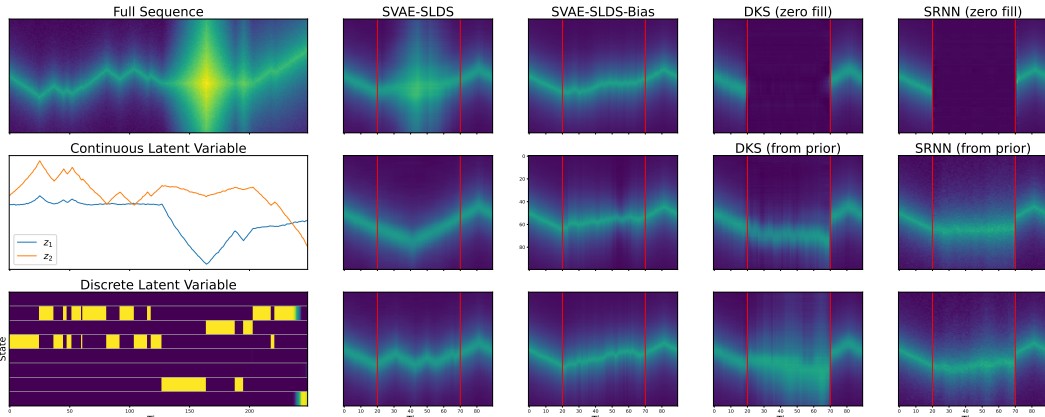

Figure 7: Segmentation and Imputation. An SLDS SVAE is trained on synthetic data with feature dimension 100, sequence length $T = 250$ and 4 distinct dynamics. We show the SVAE's latent representation of the pictured sequence via the means of the 2-dimensional continuous latent variable and the probability distributions over 8 discrete states (which has coalesced its posterior probability into a single state at nearly every time step). In the following columns we show model imputations when time steps 20 to 70 are not provided. For the DKS and SRNN, we show one imputation using the infill-with-zeros method, and 2 imputations drawn from the learned latent variable prior, conditioned on the preceding latent variables for which a posterior is properly defined.

### A.4 Discrete Deep Kalman Smoother Baselines

To our knowledge, time series VAEs that utilize Concrete [44] or straight-through gradient estimators have not been previously evaluated in literature. In order to perform a fair comparison to these approaches to gradient estimation with discrete latent variables, we have devised a new model that extends the Deep Kalman Smoother (DKS) [37] to incorporate both discrete and continuous latent variables. We can define the generative model as follows:

$$p_{\theta_k}(k_t \mid z_{t-1}, k_{t-1}) = \text{Cat}(\lambda_{\theta_k}(z_{t-1}, k_{t-1})) \tag{14}$$

$$p_{\theta_z}(z_t \mid z_{t-1}, k_t) = \mathcal{N}(\mu_{\theta_z}(z_{t-1}, k_t), \sigma_{\theta_z}) \tag{15}$$

$$p_\gamma(x_t \mid z_t) = \mathcal{N}(\mu_\gamma(z_t), \sigma_{\theta_x}) \tag{16}$$

Here $\lambda_{\theta_k}(z_{t-1}, k_{t-1})$, $\mu_{\theta_z}(z_{t-1}, k_t)$ and $\mu_\gamma(z_t)$ are neural networks, while $\sigma_{\theta_z}$ and $\sigma_{\theta_x}$ are learnable variance parameters. $\mu_{\theta_x}(z_t)$ uses the same decoder architecture as our implementation of the standard DKS model. $\lambda_{\theta_k}(z_{t-1}, k_{t-1})$, $\mu_{\theta_z}(z_{t-1}, k_t)$ use the following architecture in our experiments:

| $\lambda(\cdot)$ | $\mu(\cdot)$ |
|---|---|
| Layer Norm | Layer Norm |
| Dense 64 | Dense 64 |
| Gelu | Gelu |
| Layer Norm | Layer Norm |
| Dense 64 | Dense 64 |
| Gelu | Gelu |
| Layer Norm | Layer Norm |
| Dense | Dense |
| Softmax | |

The inference model can be written as:

$$q_{\phi_k}(k_t \mid x_{t:T}, z_{t-1}, k_{t-1}) = \text{Cat}(\lambda_{\phi_k}(g_\phi(x_{t:T}), z_{t-1}, k_{t-1})) \tag{17}$$

$$q_{\phi_z}(z_t \mid x_{t:T}, z_{t-1}, k_t) = \mathcal{N}(\mu_{\phi_z}(g_\phi(x_{t:T}), z_{t-1}, k_t), \sigma_{\phi_z}) \tag{18}$$

Here $g_\phi(x_{t:T})$ is same encoder network used by our DKS implementation (see Table 4), while $\lambda_{\phi_k}(\cdot)$ and $\mu_{\phi_k}(\cdot)$ are networks with the same architecture as $\lambda_{\theta_k}(\cdot)$ and $\mu_{\theta_k}(\cdot)$ shown above. We note that the output of $g_\phi(x_{t:T})$ is shared by both parts of the inference model.

In our experiments we match the latent dimension and number of discrete states to the corresponding SVAE SLDS model. For "DKS + Straight-Through" we use straight-through estimates of the gradient of samples of $k_t$. In this case we can directly compute the KL-divergence between $p_{\theta_k}(k_t \mid z_{t-1}, k_{t-1})$ and $q_{\phi_k}(k_t \mid x_{t:T}, z_{t-1}, k_{t-1})$ in closed form.

For "DKS + Concrete" we replace the Categorical distribution in equations 14 and 17 with a Concrete distribution. As the KL-divergence between concrete distributions does not have a closed-form expression, we estimate the KL-divergence between the two distributions by sampling. During training, we anneal the temperature parameter for each distribution from 1. to 0.1.

### A.5  Architectures and Training Specifications

All experiments use latent space with dimension $D = 16$. We train using the Adam [31] optimizer for neural network parameters, and stochastic natural gradient descent for graphical model parameters, with a batch size of $B = 128$ and learning rate $10^{-3}$ (the transformer DVAE uses learning rate $10^{-4}$, which improved performance). We train all methods for 200 epochs on MOCAP and 100 epochs on WSJ0 (including VAE-pre-training for 10 epochs for SVAE/SIN methods), which we found was sufficient for convergence.

For the SVAE-SLDS, we initialize the graphical model parameters after these 10 epochs of VAE pre-training of the networks. We found that the discrete latent variables could not meaningfully cluster the continuous latent variable at the beginning of training, when the encoder outputs do not yet contain high mutual information with the observations. After VAE pre-training, we sampled 100 sequences and encoded them. We then normalized the resulting approximate likelihood functions and sampled to obtain plausible sequences in the latent space. We trained an auto-regressive HMM model on the resulting sequences using memoized variational inference, which uses adaptive proposals to avoid local optima as implemented in bnpy [24, 25], to initialize the parameters $\eta$ of $q(\theta)$.

Table 4 summarizes the network architectures used, numbers of parameters, and compute needed to run the main experiments of this paper. Note that "Rev. LSTM" denotes a reverse-order LSTM and "TF-LSTM" refers to the "teacher-forcing" LSTM that is shared between the encoder and decoder [21]. For the transformer, attention modules in the decoder include causal masks, and all attentions are single-headed. Skip connections are not listen in the table. For a full specification of the architecture, see [42] (Table 4 outlines the architecture choices at each layer of that paper's framework).

For each model, we list the number of parameters for the MOCAP dataset. Even though the same architecture is used for both models, WSJ0 has a higher data dimension (513 vs 84) resulting in 109,824 more parameters in the encoder (via the first layer) and the decoder (in the last layer). The total amount of computation across both datasets amounts to 6 GPU days for A10G GPUs on an EC2 instance.

### A.6  Interpolation Procedure

Let $x_{\mathrm{missing}}, x_{\mathrm{obs}}$ be a partition of data into missing and observed sequences. For the following explanations, we assume that the observed data is the beginning and end of the sequence $x_{\mathrm{obs}} = x_{0.0-0.2} \cup x_{0.8-1.0}$ (first 20% and last 20% as in our results) leaving $x_{\mathrm{missing}} = x_{0.2-0.8}$.

To interpolate missing data we begin by encoding $x_{\mathrm{obs}}$ to obtain $q(z_{\mathrm{obs}})$ (in the DKS/transformer/SRNN case) or $\hat{\ell}_\phi(z_{\mathrm{obs}}|x_{\mathrm{obs}})$ (in the SVAE/SIN case), where $z_{\mathrm{obs}}$ is the subset of $z$ corresponding to the observed time steps. For models which are separable across time this is trivial, and we handle the LSTMs in the SRNN and DKS encoders by encoding every connected block of observations separately so that we do not need to hand missing or infilled data into the encoders.

For the DKS and its extensions, we then sample the remaining $z$ from the generative prior given the last encoded latent variable that precedes the missing chunk. In the regime above, we therefore sample from $p(z_{0.2-0.8}|z_{0.0-0.2})$ as we cannot easily condition on later timesteps of a forward deep recurrent model. Finally, we decode $z_{\mathrm{missing}} \cup z_{\mathrm{obs}}$ to obtain the reconstruction and interpolation.

For the SRNN and the transformer, due to their autoregressive nature we sample a value of $z$ at a single timestep (i.e. the first timestep in $z_{0.2-0.8}$) and decode to obtain a new $x$. We then repeat this

| Method | Encoder arch. | Enc. Params | Decocder arch. | Dec. Params | Latent params | Runtime |
|---|---|---|---|---|---|---|
| SVAE-SLDS | Dense 256
Gelu [22]
Layer Norm [3]
Dense 128
Gelu
Layer Norm
Dense 16 | 57,504 | Dense 128
Gelu
Layer Norm
Dense 256
Gelu
Layer Norm
Dense | 57,640 | 30,668
(50 states) | 16.4 GPU hours (MOCAP)
12.5 GPU hours (WSJ0) |
| SVAE-LDS | Dense 256
Gelu
Layer Norm
Dense 128
Gelu
Layer Norm
Dense 16 | 57,504 | Dense 128
Gelu
Layer Norm
Dense 256
Gelu
Layer Norm
Dense | 57,640 | 716 | 9 GPU hours (MOCAP)
4.6 GPU hours (WSJ0) |
| SIN-SLDS | Dense 256
Gelu
Layer Norm
Dense 128
Gelu
Layer Norm
Dense 16 | 57,504 | Dense 128
Gelu
Layer Norm
Dense 256
Gelu
Layer Norm
Dense | 57,640 | 34,018
(50 states) | 7.3 GPU hours (MOCAP)
3.6 GPU hours (WSJ0) |
| SIN-LDS | Dense 256
Gelu
Layer Norm
Dense 128
Gelu
Layer Norm
Dense 16 | 57,504 | Dense 128
Gelu
Layer Norm
Dense 256
Gelu
Layer Norm
Dense | 57,640 | 1516 | 7.3 GPU hours (MOCAP)
3.6 GPU hours (WSJ0) |
| Transformer | Dense 256
Gelu
Layer Norm
Positional Enc.
Self Attention
Layer Norm
Dense 256
Gelu + Skip
Layer Norm
Dense | 360,480 | Dense 256
Gelu
Layer Norm
Positional Enc.
Self Attention
Layer Norm
Cross Attention
Layer Norm
Dense 256
Gelu
Layer Norm
Dense | 532,136 | 518,688 | 2 GPU hours (MOCAP)
1.6 GPU hours (WSJ0) |
| SRNN | Dense 256
Gelu
Layer Norm
Dense 128
Shared LSTM 128
Reverse-LSTM 128
Dense | 252,288 | Shared LSTM 128
Dense 256
Gelu
Layer Norm
Dense | 190,888 | 87,680 | 3.7 GPU hours (MOCAP)
2.9 GPU hours (WSJ0) |
| DKS (+ Discrete) | Dense 256
Gelu
Layer Norm
Reverse-LSTM 128
Dense 192 | 244,160 | Dense 128
Gelu
Layer Norm
Dense 256
Gelu
Layer Norm
Dense | 57,640 | 38,144
(43,724) | 3.3 GPU hours (MOCAP)
2.4 GPU hours (WSJ0) |

Table 4: Summary of network architectures used for human motion capture and speech experiments. "Latent parameters" are those which specify $p(z|.)$. For the SRNN model, one LSTM layer is shared between inference and generation; we denote this layer "shared LSTM" and count its parameters among those for the decoder. SVAE-SLDS-Bias uses the same architecture as the SVAE-SLDS but takes 36.5 GPU hours MOCAP and 28.5 GPU hours on WSJ0 to train.

process so as to update the hidden state $h_t$ while holding $z_t$ in the observed and already-interpolated region constant.

For the SVAE and SIN methods, we do inference in $q_\phi(z|x)$ as in training time, except setting the inference network potentials at missing timesteps to 0 (thus setting $\hat{\ell}_\phi(z|x)$ as uniform over the reals). For models with discrete random variables the block updating routine is prone to local optima, so we initialize according to the following heuristic: we encode $x_{0.0-0.2}$ and $x_{0.8-1.0}$ and perform block updating to obtain $q(k_{0.0-0.2})$ and $q(k_{0.8-1.0})$. We then sample the endpoints $k_{0.2}, k_{0.8}$ from these variational factors. We perform HMM inference to sample from $p(k_{0.2-0.8}|k_{0.2}, k_{0.8})$, using that sample to initialize the block updating.

The protocols above can be easily extended to the $0.0 - 0.8$, $0.2 - 1.0$, and the full generation cases, noting that the models which can only condition generation forwards in time (DKS, SRNN, Transformer) must sample the initial $z$ from the generative model in the $0.0 - 0.8$ case instead of conditioning on an observation. The discrete random variable models initialize block updating for fully-generated sequences via draws from the prior $p(k)$.

Note that in Fig. 1 we show the discrete states are a discrete representation by initializing block updating with the ground truth discrete states as well as the first and last time steps (see grey background figures). We found the "anchoring" observations were necessary because the discrete states only encoder dynamics, as opposed to location in the $z$-space.

## A.7 Additional Results

In Fig. 6 we show an extended version of Fig. 5, showcasing the performance of *all* models on interpolating human motion capture data.

In Fig. 7 we compare our model's interpolation ability to several baselines on synthetic data. The data is constructed by evaluated a Laplacian distribution on a 100-dimensional grid, where observations evolve over time by either increasing or decreasing the mean of the distribution, or increasing or decreasing the variance of the distribution. Note that this data was used to test the runtimes of various methods, see Table 1.

Fig. 7 demonstrates the effect of the imputation method on the baseline models: for the RNN-based models, if we simply infill the missing observations with 0, the encoder-decoder structure simply reconstructs the 0s. Drawing the missing region from the model's prior distribution yields a generation closer to the data distribution but still infeasible under the true data dynamics. The prior SVAE implementation collapses to a single state, yielding interpolations that do not match any dynamics seen in the training data.

# B  Notation

In this section we provide a glossary of notation used in the main paper and the appendix.

## B.1  Notation in main paper

| Notation | Meaning |
|---|---|
| $\mathrm{BP}(.)$ | belief propagation function, which takes (natural) parameters of the graphical model and returns expected sufficient statistics. |
| $\gamma$ | weights of the generative network (decoder) |
| $\eta$ | natural parameters of $q(\theta; \eta)$ |
| $\tilde{\eta}$ | unconstrained parameterization of $\eta$ |
| $F_\eta$ | Fisher information matrix of exponential family distribution with natural parameters $\eta$ |
| $g(\omega; \eta, \phi, x)$ | equation which block updating attempts to solve for the root of |
| $k$ | discrete local latent variable (subset of $z$) |
| $\hat{\ell}_\phi(z\|x)$ | approximate likelihood function outputted by recognition network (encoder) |
| $\mathcal{L}$ | the loss function, the **E**vidence **L**ower **B**ound **O**bjective (ELBO). |
| $\hat{\mathcal{L}}$ | Surrogate ELBO, obtained by replacing true likelihood function with a conjugate surrogate. |
| $\mathrm{MF}(.)$ | mean field inference update, where messages are passed along residual edges (those edges removed by the structured mean field approximation). |
| $\mu$ | expected sufficient statistics of $q(z)$, i.e. $\mathbb{E}_{q(z)}[t(z)]$ |
| $\mu_m$ | expected sufficient statistics of a particular mean field component $q(z_m)$. $\mu_{-m}$ is defined analogously to $\omega_{-m}$. |
| $\mu_\eta$ | expected sufficient statistics of $q(\theta)$, i.e. $\mathbb{E}_{q(\theta)}[t(\theta)]$ |
| $\omega$ | natural parameter of $q(z)$ in any of its forms: $q(z; \eta)$ (local VI), $q_\phi(z\|x; \eta)$ (SVAE VI) or $\prod_m q_\phi(z_m\|x; \eta)$, (Structured Mean Field SVAE VI) |
| $\omega_m$ | natural parameter of each mean field cluster $\prod_m q_\phi(z_m\|x; \eta)$ in Structured Mean Field SVAE s.t. $\omega = \cup_m \omega_m$. $\omega_{-m}$ is defined analogously to $\mu_{-m}$. |
| $\omega_{-m}$ | natural parameter of all mean field clusters *but* m, $\omega_{-m} = \cup_{n \neq m} \omega_n$. |
| $\omega^*(x, \eta, \phi)$ | function which runs block updating to return the optimal natural parameters of $q(z)$ with respect to surrogate loss funciton $\hat{\mathcal{L}}$ |
| $p_0(z; \eta)$ | $p_0(z; \eta) \propto \exp\{\mathbb{E}_{q(\theta; \eta)}[\log p(z\|\theta)]\}$ |
| $\phi$ | weights of recognition network (encoder) |
| $q(\theta; \eta), q(z; \omega)$ | variational factors approximating the posteriors of global ($\theta$) and local ($z$) latent variables. |
| $q_\phi(z\|x), q_\phi(z\|x; \eta)$ | variational factor for $z$ produced by amortized VI and SVAE VI respectively. |
| $q_\phi(z_m\|x; \eta)$ | variational factor for subset $z_m \subseteq z$ |
| $t(.)$ | sufficient statistics of an exponential family distribution. |
| $\theta$ | global latent variables defining graphical model factors |
| $\mathcal{V}$ | vertex set of the graphical model |
| $x$ | observation |
| $z$ | local (per-observation) latent variable |

## B.2 Notation in this appendix

| Notation | Meaning |
|---|---|
| **Sec. C** | **Exponential families and mean field objectives** |
| $KL$ | Kullback-Leibler divergence, defined in Eq. (25) |
| $\mathcal{F}$ | set of factors in the graphical model representation of $q(z)$, introduced in Eq. (32). |
| $\lambda_\phi(x)$ | natural parameter of $\hat{\ell}_\phi(z\|x)$, defined in Eq. (44). |
| $r_t, R_t$ | for Gaussian temporal models such as the LDS and SLDS, $\lambda_\phi(x) = \{r_t, -1/2R_t\}$ as defined in Eq. (50). |
| $\theta_{\text{trans}}$ | Matrix of LDS transition parameters for each of the $K$ switching states in the SLDS, defined in Eq. (58). |
| **Sec. D** | **Belief propagation in time series** |
| $h_0, J_0$ | natural parameters of $p(z_1\|\theta)$, defined in Eq. (64). |
| $\omega_t$ | natural parameters of the LDS (or the continuous half of the SLDS) transition at time step $t$, defined in Eq. (65). Also used in the HMM as the mean field message from the LDS at step $t$. |
| $h_{1,t}, J_{11,t}, J_{12,t}, J_{22,t}, h_{2,t}$ | components of $\omega_t$, defined in Eq. (65). |
| $m_{z_i,\omega_i}(z_i)$ | belief propagation messages from variable $z_i$ to factor with parameter $\omega_i$, introduced in Eq. (69) |
| $m_{\omega_i,z_i}(z_i)$ | belief propagation messages from factor with parameter $\omega_i$ to variable $z_i$, introduced in Eq. (70) |
| $f_{t\|t}, F_{t\|t}$ | parameters of $m_{z_t,\omega_{t+1}}(z_t)$, i.e. the *filtered* messages for the Gaussian inference at step $t$ which has accrued evidence from the first $t$ "observations" |
| $f_{t+1\|t}, F_{t+1\|t}$ | parameters of $m_{\omega_{t+1},z_{t+1}}(z_{t+1})$, i.e. the *predicted* messages for the Gaussian inference at step $t+1$ which has accrued evidence from the first $t$ "observations" |
| $f_{t\|T}, F_{t\|T}$ | parameters of $q(z_t)$, i.e. the *smoothed* messages for the Gaussian inference at step $t$ which has accrued evidence from all "observations" |
| $\text{ker}(.)$ | returns an un-normalized distribution with no log partition function, defined in Eq. (111) |
| $f_t(z_{t-1}, z_t), \phi_{\cdot,t}, \Phi_{\cdot\cdot,t}$ | sequence of functions (and their parameters) used by the filtering step of the parallel Kalman filter, defined in Eq. (115) |
| $g_t(z_{t-1}), \gamma_t, \Gamma_t$ | sequence of functions (and their parameters) used by the filtering step of the parallel Kalman filter, defined in Eq. (112) |
| $e_t(z_t, z_{t+1}), \epsilon_t, E_t$ | sequence of functions (and their parameters) used by the smoothing step of the parallel Kalman smoother, defined in Eq. (148) |
| $\otimes$ | associative operator for parallel Kalman filtering, defined in Eq. (119) and (120) |
| $\oplus$ | associative operator for parallel Kalman smoothing, defined in Eq. (151) |

# C  Exponential families and mean field objective

To rigorously define the SVAE inference routine, in Sec. C.1 we (re-)introduce exponential family and factor graph notation for defining distributions. Then, in Sec. C.2 we derive the optimizer of $\hat{\mathcal{L}}[\dots]$ and the mean field message passing update $\mathtt{MF}(.)$. Sec. D then contains the necessary equations for belief propagation in the LDS and SLDS models, including proofs and derivations of our novel inference algorithm which parallelizes belief propagation across time steps.

## C.1  Exponential families of distributions

Given an observation $z \in \mathcal{Z}$, an exponential family is a family of distributions characterized by a statistic function $t(z) : \mathcal{Z} \to \mathbb{R}^n$ with probabilities given by

$$p(z; \eta) = \exp\{\langle \eta, t(z) \rangle - \log Z(\eta)\} \tag{19}$$

Where $\eta \in \mathbb{R}^n$ is a parameter vector known as the *natural parameters*, and the *log partition function* $\log Z(\eta)$ is given by

$$\log Z(\eta) = \log \int_z \exp\{\langle \eta, t(z) \rangle\} dx \tag{20}$$

and ensures that the distribution properly normalizes (to simplify notation, here we focus on continuous $z$ but this definition can be easily extended to discrete observations by replacing the integral with a sum and and restricting the output of $t(z)$ to some discrete set). A single assignment of values to $\eta$ characterizes a member of this exponential family, and so searching over a family means searching over valid values of $\eta \in H$ where $H$ is the set of parameters for which the integral in Eq. (20) is finite. We begin with a couple simple properties of exponential families:

**Proposition C.1.** *The gradient of a distribution's log partition function with respect to its parameters is its expected statistics:*

$$\nabla_\eta \log Z(\eta)) = \mathbb{E}_{p(z;\eta)}[t(z)] \tag{21}$$

*Proof.*

$$\nabla_\eta \log Z(\eta) = \frac{1}{\int_z \exp\{\langle \eta, t(z) \rangle\} dx} \int_z t(z) \exp\{\langle \eta, t(z) \rangle\} dx \tag{22}$$

$$= \int_z t(z) \exp\{\langle \eta, t(z) \rangle - \log Z(\eta)\} = \mathbb{E}_{p(z;\eta)}[t(z)] \tag{23}$$

$\square$

**Proposition C.2.** *Given two distributions $p(z; \eta_1), p(z; \eta_2)$ in the same exponential family with parameters $\eta_1$ and $\eta_2$, the KL divergence between the two distributions is given by*

$$\mathrm{KL}(p(z; \eta_1) \parallel p(z; \eta_2))) := \mathbb{E}_{p(z;\eta_1)}\left[ \log \frac{p(z; \eta_1)}{p(z; \eta_2)} \right] \tag{24}$$

$$= \langle \eta_1 - \eta_2, \mathbb{E}_{p(z;\eta_1)}[t(z)] \rangle - \log Z(\eta_1) + \log Z(\eta_2) \tag{25}$$

*Proof.* The log probabilities are linear in the statistics, so the expectation can be pulled in. $\square$

**Definition C.3.** Given an exponential family distribution $p(\theta; \eta)$

$$p(\theta; \eta) = \exp\{\langle \eta, t(\theta) \rangle - \log Z(\eta)\} \tag{26}$$

we say that $p(z|\theta)$ is **conjugate** to $p(\theta; \eta)$ if one distribution's parameters is the other distribution's statistics, namely

$$p(z|\theta) = \exp\{\langle t(\theta), (t(z), 1) \rangle - c\} = \exp\{\langle \hat{t}(\theta), t(z) \rangle - \log Z(\hat{t}(\theta))\} \tag{27}$$

Where $\hat{t}(\theta)$ is the first $\dim(t(z))$ entries of $t(\theta)$ and $c$ is a constant.

Note that $t(\theta)$ and $t(z)$ are not the same function with different inputs, but rather are unique to each distribution. As it is generally clear which function is needed, we share this notation, just as $p(\theta), p(z)$ is understood to refer to different densities.

Further note that the sufficient statistics of $p(\theta)$ either become parameters of $p(z|\theta)$ or its normalizers. This can be notated two ways: either by appending 1s to the end of $t(z)$ to align with the normalizers, or simply fold them into the log partition function. An example is provided below.

**Example: NIW distribution.** A normal distribution $p(z; \mu, \Sigma)$ has natural parameters and sufficient statistics given by:

$$\eta = \begin{bmatrix} \Sigma^{-1}\mu \\ -\frac{1}{2}\Sigma^{-1} \end{bmatrix} \qquad t(x) = \begin{bmatrix} x \\ xx^T \end{bmatrix} \tag{28}$$

We bend notation here by concatenating vectors to matrices. Assume all matrices are appropriately flattened to make the inner product a scalar, for more info see Sec. G. The normal distribution's log partition function is given by:

$$\log Z(\eta) = \frac{1}{2}\mu^T \Sigma^{-1}\mu + \frac{1}{2}\log|\Sigma| + \frac{n}{2}\log(2\pi) \tag{29}$$

Meanwhile, the Normal Inverse Wishart (NIW) distribution has sufficient statistics:

$$t(\mu, \Sigma) = \begin{bmatrix} -\frac{1}{2}\Sigma^{-1} \\ \Sigma^{-1}\mu \\ -\frac{1}{2}\mu^T\Sigma^{-1}\mu \\ -\frac{1}{2}\log|\Sigma| \end{bmatrix} \tag{30}$$

Note that the first two are parameters of $p(z)$, and the last two appear in the log partition function. A desirable property of conjugacy is that the posterior has a simple form, namely:

$$p(\theta|z) \propto \exp\{\langle t(\theta), \eta + (t(z), 1)\rangle\} \tag{31}$$

This follows easily from the fact that $p(\theta|z) \propto p(\theta)p(z|\theta)$ which are both log-linear in $t(\theta)$.

**Factor Graphs.** The exponential family framework can be applied to complicated distributions over many variables, in which case it can be helpful to rephrase the distribution as a collection of factors on subsets of the variables. We represent the variable $z$ as a graphical model with vertex set $\mathcal{V}$ such that the vertices partition the components of $z$. Let the *factor set* $\mathcal{F}$ be a set of subsets of $\mathcal{V}$ such that $z_f \subseteq z, f \in \mathcal{F}$. Then we write our distribution

$$p(z; \eta) = \exp\{\langle \eta, t(z)\rangle - \log Z(\eta)\} = \exp\left\{\sum_{f \in \mathcal{F}} \langle \eta_f, t(z_f)\rangle - \log Z(\eta)\right\} \tag{32}$$

These two ways of writing the distribution are trivially equal by defining $\eta$ as the union of all $\eta_f$ and combining $t(z_f)$ appropriately. We demonstrate where this notation can be useful in two examples.

## C.2 Mean field Optima

**Proposition C.4.** *Given a fixed distribution $q(a)$ and a loss function of the form*

$$l[q(a)q(b)] = \mathbb{E}_{q(a)q(b)}\left[\log \frac{f(a,b)}{q(a)q(b)}\right] \tag{33}$$

*for some non-negative, continuous function $f(a, b)$ with finite integral (such as an un-normalized density), the partially optimal $q(b)$ across all possible distributions is given by*

$$\underset{q(b)}{\arg\max}\, l[q(a)q(b)] \propto \exp\{\mathbb{E}_{q(a)}[\log f(a,b)]\} \tag{34}$$

*Proof.* This is a simple extension of a proof in Johnson et al. [29]. Let $\tilde{p}(b)$ be the distribution which satisfies $\tilde{p}(b) \propto \exp\{\mathbb{E}_{q(a)}[\log f(a,b)]\}$. We can drop terms in the loss which are constant with respect to $q(b)$ and write:

$$\underset{q(b)}{\arg\max}\, \mathbb{E}_{q(a)q(b)}\left[\log \frac{f(a,b)}{q(b)}\right] = \underset{q(b)}{\arg\max}\, \mathbb{E}_{q(b)}[\mathbb{E}_{q(a)}[\log f(a,b)] - \log q(b)] \tag{35}$$

$$= \underset{q(b)}{\arg\max}\, \mathbb{E}_{q(b)}[\log \exp \mathbb{E}_{q(a)}[\log f(a,b)] - \log q(b)] \tag{36}$$

$$= \underset{q(b)}{\arg\max}\, \mathbb{E}_{q(b)}\left[\log \frac{\exp\{\mathbb{E}_{q(a)}[\log f(a,b)]\}}{q(b)}\right] \tag{37}$$

$$= \underset{q(b)}{\arg\max}\, \mathbb{E}_{q(b)}\left[\log \frac{\tilde{p}(b)}{q(b)} + \text{const}\right] \tag{38}$$

$$= \underset{q(b)}{\arg\max}\, -\text{KL}(q(b) \| \tilde{p}(b)) = \tilde{p}(b) \tag{39}$$

$\square$

This is applicable to ELBO maximization problems, as well as the surrogate maximization problem arising in the SVAE:

**Example: Exact Inference in the SVAE**  Applying proposition C.4 to the SVAE's surrogate objective, we see:

$$\underset{q(z)}{\arg\max}\,\hat{\mathcal{L}}[q(\theta;\eta)q(z);\phi] = \mathbb{E}_{q(\theta;\eta)q(z)}\left[\log\frac{p(\theta)p(z|\theta)\hat{\ell}_\phi(z|x)}{q(\theta;\eta)q(z)}\right] \tag{40}$$

$$\propto \exp\{\mathbb{E}_{q(\theta;\eta)}[\log p(\theta)\cdot p(z|\theta)\cdot\hat{\ell}_\phi(z|x)]\} \tag{41}$$

$$\propto \exp\{\mathbb{E}_{q(\theta;\eta)}[\log p(\theta)] + \mathbb{E}_{q(\theta;\eta)}[\log p(z|\theta)] + \log\hat{\ell}_\phi(z|x)\} \tag{42}$$

$$\propto \exp\{\mathbb{E}_{q(\theta;\eta)}[\log p(z|\theta)] + \log\hat{\ell}_\phi(z|x)\} \tag{43}$$

Recall that we have chosen $\hat{\ell}_\phi(z|x)$ to be *a conjugate likelihood function* to $p(z|\theta)$. Let $\lambda_\phi(x)$ be the outputs of the recognition network such that

$$\hat{\ell}_\phi(z|x) = \exp\{\langle\lambda_\phi(x), t(z)\rangle\} \tag{44}$$

Then

$$\underset{q(z)}{\arg\max}\,\hat{\mathcal{L}}[q(\theta;\eta)q(z);\phi] \propto \exp\{\mathbb{E}_{q(\theta;\eta)}[\log p(z|\theta)] + \log\hat{\ell}_\phi(z|x)\} \tag{45}$$

$$\propto \exp\{\mathbb{E}_{q(\theta;\eta)}[\langle t(\theta), t(z)\rangle] + \langle\lambda_\phi(x), t(z)\rangle\} \tag{46}$$

$$= \exp\{\langle\mathbb{E}_{q(\theta;\eta)}[t(\theta)] + \lambda_\phi(x), t(z)\rangle\} \tag{47}$$

Demonstrating that the optimal $q(z)$ is in the same exponential family as $p(z|\theta)$, with parameters $\omega = \mathbb{E}_{q(\theta;\eta)}[t(\theta)] + \lambda_\phi(x)$. These parameters are the sum of *expected* parameters from the prior and the parameters of the fake observations.

**Example: Linear Dynamical System.**  Consider the linear dynamical system, where $p(z_1) = \mathcal{N}(\mu_0, \Sigma_0)$ and $p(z_t|z_{t-1}) = \mathcal{N}(A_t z_{t-1} + b_t, \Sigma_t)$ (in practice, we use a time-homogenous model in which $A, b, \Sigma$ do not depend on the time step $t$). We can write this distribution

$$p(z|\theta) = p(z_1)\prod_{t=2}^{T}p(z_t|z_{t-1}) = \exp\left\{\langle t(\theta_1), t(z_1)\rangle + \sum_{t=2}^{T}\langle t(\theta_t), t(z_{t-1}, z_t)\rangle - \log Z(t(\theta))\right\} \tag{48}$$

Where $t(z_1), t(\theta_1)$ are given by the NIW example in Sec. C.1, and for $t = 2, \ldots, T$,

$$t(z_{t-1}, z_t) = \begin{bmatrix} z_{t-1} \\ z_{t-1}z_{t-1}^T \\ z_{t-1}z_t^T \\ z_t z_t^T \\ z_t \end{bmatrix}, \quad t(\theta_t) = \begin{bmatrix} -A_t^T Q_t^{-1} b_t \\ -\frac{1}{2}A_t^T Q_t^{-1} A_t \\ A_t^T Q_t^{-1} \\ -\frac{1}{2}Q_t^{-1} \\ Q_t^{-1}b_t \end{bmatrix} \tag{49}$$

Note that both $t(\theta_t)$ and $t(\theta_{t+1})$ introduce parameters corresponding to the statistics $z_t$. In the context of the SVAE, we add observations to each time step,

$$\hat{\ell}_\phi(z|x) = \exp\left\{\sum_t\left\langle\begin{bmatrix} r_t \\ -\frac{1}{2}R_t \end{bmatrix}, \begin{bmatrix} z_t \\ z_t z_t^T \end{bmatrix}\right\rangle\right\} \tag{50}$$

which is conjugate to the likelihood $p(z|\theta)$ so that the mean field optimum stays in the same exponential family.

**Structured mean field.**  The main paper introduces the concept of dividing the latent variables into separate structured mean field components $z_m \subset z, m \in \mathcal{M}$. We outline the math in this case by noting that conjugacy allows us to break down statistics on groups of vertices into statistics on individual vertices.

$$\langle t(\theta), t(z_{t-1}, z_t)\rangle = \langle t(\theta, z_{t-1}), t(z_t)\rangle = \langle t(\theta, z_t), t(z_{t-1})\rangle \tag{51}$$

i.e. this expression is linear in the statistics of *each* argument. This proves important in deriving the optimal factor $q(z_m)$ given the other factors $q(z_{-m})$. By applying prop C.4 to the mean field surrogate objective with $q(a) = q(\theta) \prod_{-m} q(z_m)$ and $q(b) = q(z_m)$, we see:

$$\underset{q(z_m)}{\arg\max} \hat{\mathcal{L}}[q(\theta; \eta) \prod_m q(z_m); \phi] \propto \exp\{\mathbb{E}_{q(\theta;\eta)q(z_{-m})}[\log p(z|\theta)] + \log \hat{\ell}_\phi(z|x)\} \qquad (52)$$

The expected-prior part of this expression can be rewritten (where $z_{f,m} = z_f \cap z_m$ and $z_{f,-m} = z_f \cap z_m^C$):

$$\mathbb{E}_{q(\theta;\eta)q(z_{-m})}[\log p(z|\theta)] \propto \mathbb{E}_{q(\theta;\eta)q(z_{-m})}\Big[ \sum_{f \in \mathcal{F}} \langle t(\theta_f), t(z_f) \rangle \Big] \qquad (53)$$

$$= \mathbb{E}_{q(\theta;\eta)q(z_{-m})}\Big[ \sum_{f \in \mathcal{F}} \langle t(\theta_f, z_{f,-m}), t(z_{f,m}) \rangle \Big] \qquad (54)$$

$$= \sum_{f \in \mathcal{F}} \langle \mathbb{E}_{q(\theta;\eta)q(z_{-m})}[t(\theta_f, z_{f,-m})], t(z_{f,m}) \rangle \qquad (55)$$

Note that because the joint sufficient statistics of $z_{f,-m}, \theta_f$ are linear in each input, we can distribute the expectations through the statistic function and write

$$\mathbb{E}_{q(\theta;\eta)q(z_{-m})}[t(\theta_f, z_{f,-m})] = \mathtt{MF}(\mathbb{E}_{q(\theta;\eta)}[t(\theta_f)], \mathbb{E}_{q(z_{-m})}[t(z_{f,-m})]) \qquad (56)$$

Where $\mathtt{MF}(.)$ is a mean field function which takes the statistics of $\theta_f, z_{f,-m}$ and produces the joint statistics $t(\theta_f, z_{f,-m})$. This function is linear in each of its inputs and generally cheap to compute.

**Example: Switching Linear Dynamical System.** The SLDS consists of the following generative model: $p(k_2) = \mathrm{Cat}(\pi_0)$, $p(k_t) = \mathrm{Cat}(\pi_{k_{t-1}})$, $p(z_1) = \mathcal{N}(\mu_0, \Sigma_0)$, $p(k_2) = \mathrm{Cat}(\pi_0)$ and $p(z_t|z_{t-1}) = \mathcal{N}(A_{k_t} z_{t-1} + b_{k_t}, \Sigma_{k_t})$.

In an exponential family form, the model consists of the following factors: (i) an initial-state factor on $z_1$ matching the NIW example in Eqs. (10-12), (ii) an initial categorical factor on $k_2$ (see Sec. G for the form), (iii) a transition factor for $p(k_t|k_{t-1})$ which has parameters $\pi_{ij}$ and sufficient statistics $1\{(k_t = i)\&(k_{t+1} = j)\}$ for $i, j = 1, \ldots, K$, and (iv) transition factors corresponding to $p(z_t|z_{t-1}, k_t)$, which we will examine further.

Recall from Eq. (49) the expression for the parameters of $p(z_t|z_{t-1})$ in the LDS. Then we can write:

$$p(z_t|z_{t-1}, k_t) = \exp\{\langle t(\theta_{k_t}), t(z_{t-1}, z_t) \rangle - \log Z(t(\theta_{k_t}))\} \qquad (57)$$

We need to rewrite this to be an exponential family distribution of $k_t$ as well (instead of only appearing as an index). Note that the sufficient statistics of a categorical variable is simply its one-hot encoding. If we define $t(\theta_{\mathrm{trans}})$ to be a matrix whose $k$th row is given by $t(\theta_k)$, and $\log Z(\theta_{\mathrm{trans}})$ to be a vector with components $\log Z(\theta_k)$, we write:

$$p(z_t|z_{t-1}, k_t) = \exp\{\langle t(\theta_{\mathrm{trans}}), t(k_t) \cdot t(z_{t-1}, z_t)^T \rangle + \langle -\log Z(t(\theta_{\mathrm{trans}})), t(k_t) \rangle\} \qquad (58)$$

Thus the conditional distribution can be described by two factors, one which takes the outer product of the one-hot encoding $t(k_t)$ with the continuous observations (thus embedding them in the correct dimension for inner product with $t(\theta)$), and one which simply takes the correct log normalizer based on what state we're in (again, by multiplying a one-hot encoding with a vector of log probabilities).

Given observations, exact inference in the SLDS is intractable. Thus we divide the graphical model via structured mean field inference into $q(z)q(k)$. This separates out the joint factors $\langle t(\theta_{\mathrm{trans}}), t(k_t) \cdot t(z_{t-1}, z_t)^T \rangle$. However, as noted in Eq. (51) we can reframe this as a factor on each mean field cluster:

$$\exp\{\langle t(\theta_{\mathrm{trans}}), t(k_t) \cdot t(z_{t-1}, z_t)^T \rangle\} = \exp\{\langle t(z_{t-1}, z_t), t(\theta_{\mathrm{trans}})^T \cdot t(k_t) \rangle\} \qquad (59)$$

$$= \exp\{\langle t(k_t), t(\theta_{\mathrm{trans}}) \cdot t(z_{t-1}, z_t) \rangle\} \qquad (60)$$

Thus, applying Eq. (55), the optimal $q(k)$ inherits the (expected) initial-state factors from $p(k_2)$ and transition factors from $p(k_t|k_{t-1})$, as well as the *normalizer* factors $\langle -\log Z(t(\theta_{\mathrm{trans}})), t(k_t) \rangle$ from Eq. (58), but replaces the joint factors of Eq. (58) which include $z_{t-1}, z_t$ with

$$\langle \mathbb{E}_{q(\theta;\eta)}[t(\theta_{\mathrm{trans}})] \cdot \mathbb{E}_{q(z)}[t(z_{t-1}, z_t)], t(k_t) \rangle \qquad (61)$$

Which (when combined with the normalizer factor) is simply a vector of expected log probabilities that the transition at time $t$ is caused by each state. Similarly, the continuous chain keeps its initial state factor corresponding to $p(z_1)$ but all of its transition factors are replaced by

$$\langle \mathbb{E}_{q(\theta;\eta)}[t(\theta_{\mathrm{trans}})]^T \cdot \mathbb{E}_{q(k_t)}[t(k_t)], t(z_{t-1}, z_t) \rangle \qquad (62)$$

Because $t(k_t)$ is a one-hot encoding of $k_t$, $\mathbb{E}_{q(k_t)}[t(k_t)]$ is simply a probability vector of what value $k_t$ takes. Thus the new parameter for $t(z_{t-1}, z_t)$ is simply a convex combination of the rows of $t(\theta_{\text{trans}})$, yielding a time-inhomogeneous LDS with each transition computed as an expectation over discrete states. Our mean field optimization of the SLDS becomes:

- Run a belief propagation algorithm to get $\mathbb{E}_{q(z)}[t(z)]$
- For each $t$, compute $\mathbb{E}_{q(\theta)}[t(\theta_{\text{trans}})] \cdot \mathbb{E}_{q(z)}[t(z_{t-1}, z_t)]$ as the new parameters in $q(k)$ corresponding to $t(k_t)$
- Run a belief propagation algorithm to get $\mathbb{E}_{q(k)}[t(k)]$
- For each $t$, compute $\mathbb{E}_{q(\theta)}[t(\theta_{\text{trans}})]^T \cdot \mathbb{E}_{q(k)}[t(k_t)]$ as the new parameters in $q(z)$ corresponding to $t(z_{t-1}, z_t)$

. . . and repeat until convergence.

## D  Belief propagation in time series

### D.1  Sequential LDS Inference

Consider again the generative model
$$z_1 \sim \mathcal{N}(\mu_0, \Sigma_0), \quad z_t \sim \mathcal{N}(A_t z_{t-1} + b_t, Q_t) \tag{63}$$
Recall from Eq. (47) that the $q(z; \omega)$ will have some parameters from the expected log prior and some parameters from the recognition potential. We separate out the parameters into individual factors, starting with a prior factor on the initial state:
$$\omega_0 = \begin{bmatrix} h_0 \\ -\frac{1}{2} J_0 \end{bmatrix} \quad t(z_1) = \begin{bmatrix} z_1 \\ z_1 z_1^T \end{bmatrix} \tag{64}$$
Where $h_0, -\frac{1}{2} J_0$ are the expected statistics of an NIW prior. Further, we place a transition factor on each adjacent pair of states, given by Eq. (49). To simplify notation, we write
$$t(z_t, z_{t+1}) = \begin{bmatrix} z_t \\ z_t z_t^T \\ z_t z_{t+1}^T \\ z_{t+1} z_{t+1}^T \\ z_{t+1} \end{bmatrix}, \quad \omega_{t+1} = \begin{bmatrix} -h_{1,t} \\ -\frac{1}{2} J_{11,t} \\ J_{12,t} \\ -\frac{1}{2} J_{22,t} \\ h_{2,t} \end{bmatrix} = \begin{bmatrix} -\mathbb{E}[A_t^T Q_t^{-1} b_t] \\ -\frac{1}{2} \mathbb{E}[A_t^T Q_t^{-1} A_t] \\ \mathbb{E}[A_t^T Q_t^{-1}] \\ -\frac{1}{2} \mathbb{E}[Q_t^{-1}] \\ \mathbb{E}[Q_t^{-1} b_t] \end{bmatrix} \tag{65}$$
For the SLDS, each of these parameters is outputted by the mean field message passing function and is a convex combination of the corresponding parameters for each cluster, weighted by $\mathbb{E}_{q(k_t)}[t(k_t)]$. For the LDS, these values are time-homogeneous.

Note that there is also a factor outputted by the recognition network output at each time step:
$$\lambda_t = \begin{bmatrix} r_t \\ -\frac{1}{2} R_t \end{bmatrix} \quad t(z_t) = \begin{bmatrix} z_t \\ z_t z_t^T \end{bmatrix} \tag{66}$$
Because the parameters are expectations, there may be no values of $\hat{A}_t, \hat{Q}_t, \hat{b}_t$ which satisfy
$$\begin{bmatrix} -\hat{A}_t^T \hat{Q}_t^{-1} \hat{b}_t \\ -\frac{1}{2} \hat{A}_t^T \hat{Q}_t^{-1} \hat{A}_t \\ \hat{A}_t^T \hat{Q}_t^{-1} \\ -\frac{1}{2} \hat{Q}_t^{-1} \\ \hat{Q}_t^{-1} \hat{b}_t \end{bmatrix} = \begin{bmatrix} -h_{1,t} \\ -\frac{1}{2} J_{11,t} \\ J_{12,t} \\ -\frac{1}{2} J_{22,t} \\ h_{2,t} \end{bmatrix} \tag{67}$$
And as a result, it might not be equivalent to any single LDS with fixed parameters. This prevents us from using general Kalman smoother algorithms, so we must return to the definition of exponential families and belief propagation.

The goal of inference is to compute the marginals $q(z_t)$, draw samples, and compute $\log Z$. Using the belief propagation algorithm, we know $q(z_t)$ is proportional to the product of incoming *messages* from factors which involve $z_t$. In belief propagation, messages pass from *factors* to *variables* and back. Thus we write
$$q(z_t) \propto m_{\omega_t, z_t}(z_t) \cdot m_{\omega_{t+1}, z_t}(z_t) \cdot m_{\lambda_t, z_t}(z_t) \tag{68}$$

Where $m_{\omega,z}(z)$ is the message from the factor with parameter $\omega$ to variable $z$. We similarly define $m_{z,\omega}(z)$ to be the message from variable to factor. Note that all $m_{.,.}(z_t)$ are provably Gaussian, and can therefore each message function can be minimally described by the Gaussian natural parameters.

Messages from variables to factors are the product of incoming messages, e.g.

$$m_{z_t,\omega_{t+1}}(z_t) = m_{\omega_t,z_t}(z_t) \cdot m_{\lambda_t,z_t}(z_t) \tag{69}$$

While messages from factors to variables are computed by integrating over incoming messages times the factor potential, e.g.

$$m_{\omega_t,z_t}(z_t) = \int_{z_{t-1}} m_{z_{t-1},\omega_t}(z_{t-1}) \cdot \exp\{\langle \omega_t, t(z_{t-1}, z_t) \rangle\} dz_{t-1} \tag{70}$$

We attempt to match our notation to the typical Kalman smoother algorithm, by defining *filtered messages* $J_{t|t}, h_{t|t}$ as the natural parameters of $m_{z_t,\omega_{t+1}}(z_t)$, i.e. the distribution which has accumulated the first $\omega_{1:t}$ and $\lambda_{1:t}$, which correspond to the first $t$ "observations". We similarly define *predicted messages* $J_{t+1|f}, h_{t+1|f}$ as the natural parameters of $m_{\omega_{t+1},z_{t+1}}(z_{t+1})$, which corresponds to a distribution over $z_{t+1}$ given the first $t$ "observations". Finally we describe the smoothed messages $J_{t|T}, h_{t|T}$ to be the natural parameters of $q(z_t)$ (note that for simplicity the natural parameters for each distribution are actually $-\frac{1}{2}J_{.|.}, h_{.|.}$ and the negative one half is assumed to be factored out).

The message passing algorithm is outlined below:
**Input:** $\omega_t, \lambda_t$.
**Initialize**:

$$F_{1|1} = J_0 + R_1 \tag{71}$$
$$f_{1|1} = h_0 + r_1 \tag{72}$$
$$\log Z = 0 \tag{73}$$

**Filter:** for $t = 1, \ldots, T-1$
Predict:

$$P_t = F_{t|t} + J_{11,t} \tag{74}$$
$$F_{t+1|t} = J_{22,t} - J_{12,t}^T \cdot P_t^{-1} \cdot J_{12,t} \tag{75}$$
$$f_{t+1|t} = h_{2,t} + J_{12,t}^T \cdot P_t^{-1} \cdot (f_{t|t} - h_{1,t}) \tag{76}$$
$$\log Z = \log Z + \frac{1}{2}\left((f_{t|t} - h_{1,t})^T P_t^{-1}(f_{t|t} - h_{1,t}) - \log|P_t|\right) \tag{77}$$

Measurement:

$$F_{t+1|t+1} = F_{t+1|t} + R_{t+1} \tag{78}$$
$$f_{t+1|t+1} = f_{t+1|t} + r_{t+1} \tag{79}$$

**Finalize:**

$$\log Z = \log Z + \frac{1}{2}\left(f_{T|T}^T F_{T|T}^{-1} F_{T|T} - \log|F_{T|T}|\right) \tag{80}$$

**Smoother**: for $t = T-1, \ldots, 1$

$$C_t = F_{t+1|T} - F_{t+1|t} + J_{22,t} \tag{81}$$
$$F_{t|T} = F_{t|t} + J_{11,t} - J_{12,t} \cdot C_t^{-1} \cdot J_{12,t}^T \tag{82}$$
$$f_{t|T} = f_{t|t} - h_{1,t} + J_{12,t}C_t^{-1}(f_{t+1|T} - f_{t+1|t} + h_{2,t}) \tag{83}$$

**Sample:** (the $\mathcal{N}$ below are taking natural parameters as input)

$$z_T \sim \mathcal{N}\left(f_{T|T}, -\frac{1}{2}F_{T|T}\right) \tag{84}$$

$$\texttt{for } t = T-1, \ldots, 1 \quad z_t \sim \mathcal{N}\left(f_{t|t} + J_{12,t}z_{t+1} - h_{1,t}, -\frac{1}{2}(F_{t|t} + J_{11,t})\right) \tag{85}$$

**Return:** marginal params $f_{t|T}, F_{t|T}$; $\log Z$; and samples $x_{1:T}$.

From the log marginals, we can recover the expected statistics:

$$\Sigma_i = F_{t|T}^{-1} \tag{86}$$

$$\mu_i = \Sigma_i f_{t|T} \tag{87}$$

$$E[z_i] = \mu_i \tag{88}$$

$$E[z_i z_i^T] = \Sigma_i + \mu_i \mu_i^T \tag{89}$$

$$E[z_i z_{i+1}^T] = \Sigma_i J_{12,t} C_t^{-1} + \mu_i \mu_{i+1}^T \tag{90}$$

**Derivation of filter:** The measurement step trivially follows from Eq. (69). The predict step, on the other hand, attempts to compute Eq. (70), which can be written out:

$$\int_{z_t} \exp\left\{ \left\langle \begin{bmatrix} f_{t|t} \\ -\frac{1}{2}F_{t|t} \end{bmatrix}, \begin{bmatrix} z_t \\ z_t z_t^T \end{bmatrix} \right\rangle + \left\langle \begin{bmatrix} -\frac{1}{2}J_{11,t} \\ J_{12,t} \\ -\frac{1}{2}J_{22,t} \\ -h_{1,t} \\ h_{2,t} \end{bmatrix}, \begin{bmatrix} z_t z_t^T \\ z_t z_{t+1}^T \\ z_{t+1} z_{t+1}^T \\ z_t \\ z_{t+1} \end{bmatrix} \right\rangle \right\} dz_t \tag{91}$$

$$= \exp\left\{ \left\langle \begin{bmatrix} -\frac{1}{2}J_{22,t} \\ h_{2,t} \end{bmatrix}, \begin{bmatrix} z_{t+1} z_{t+1}^T \\ z_{t+1} \end{bmatrix} \right\rangle \right\} \int_{z_t} \exp\left\{ \left\langle \begin{bmatrix} f_{t|t} + J_{12,t} z_{t+1} - h_{1,t} \\ -\frac{1}{2}(F_{t|t} + J_{11,t}) \end{bmatrix}, \begin{bmatrix} z_t \\ z_t z_t^T \end{bmatrix} \right\rangle \right\} dz_t \tag{92}$$

$$= \exp\left\{ \left\langle \begin{bmatrix} -\frac{1}{2}J_{22,t} \\ h_{2,t} \end{bmatrix}, \begin{bmatrix} z_{t+1} z_{t+1}^T \\ z_{t+1} \end{bmatrix} \right\rangle + \log Z(f_{t|t} - h_{1,t} + J_{12,t} z_{t+1}, F_{t|t} + J_{11,t}) \right\} \tag{93}$$

What is that log partition function term at the end?

$$\frac{1}{2}(f_{t|t} - h_{1,t} + J_{12,t} z_{t+1})^T (F_{t|t} + J_{11,t})^{-1}(f_{t|t} - h_{1,t} + J_{12,t} z_{t+1}) - \frac{1}{2}\log|F_{t|t} + J_{11,t}| \tag{94}$$

Simplifying $P_t = F_{t|t} + J_{11,t}$, we recover normalization term

$$\frac{1}{2}(f_{t|t} - h_{1,t})^T P_t^{-1}(f_{t|t} - h_{1,t}) - \frac{1}{2}\log|P_t| \tag{95}$$

Linear term

$$z_{t+1}^T J_{12,t}^T P_t^{-1}(f_{t|t} - h_{1,t}) \tag{96}$$

And quadratic term

$$\frac{1}{2}z_{t+1}^T J_{12,t}^T P_t^{-1} J_{12,t} z_{t+1} \tag{97}$$

Thus the entire message is:

$$\exp\left\{ \left\langle \begin{bmatrix} h_{2,t} + J_{12,t}^T P_t^{-1}(f_{t|t} - h_{1,t}) \\ -\frac{1}{2}(J_{22,t} - J_{12,t}^T P_t^{-1} J_{12,t}) \end{bmatrix}, \begin{bmatrix} z_{t+1} \\ z_{t+1} z_{t+1}^T \end{bmatrix} \right\rangle + \frac{1}{2}(f_{t|t} - h_{1,t})^T P_t^{-1}(f_{t|t} - h_{1,t}) - \frac{1}{2}\log|P_t| \right\} \tag{98}$$

To calculate $\log Z$, instead of normalizing each message to be a valid distribution, we can think of normalizing each message so it's of the form $\exp\{\langle \eta, t(z)\rangle\}$, i.e. remove all normalizers. Then to calculate the entire model's $\log Z$, we must accrue all normalizers that appear in predict steps (none appear in measurement steps) and simply add the normalizer for the last message.

**Derivation of Smoother:** Note that

$$q(z_t) \propto m_{\omega_{t+1}, z_t}(z_t) \cdot m_{z_t, \omega_{t+1}}(z_t) \tag{99}$$

The second term is the filtered message computed in the filtering step. Thus we must simply compute the backwards messages. The message becomes:

$$q(z_t) \propto m_{z_t, \omega_{t+1}}(z_t) \cdot m_{\omega_{t+1}, z_t}(z_t) \tag{100}$$

$$\propto m_{z_t, \omega_{t+1}}(z_t) \cdot \int m_{z_{t+1}, \omega_{t+1}}(z_{t+1}) \exp\{\langle \omega_{t+1}, t(z_t, z_{t+1})\rangle\} dz_{t+1} \tag{101}$$

$$= m_{z_t, \omega_{t+1}}(z_t) \cdot \int (m_{\omega_{t+2}, z_{t+1}}(z_{t+1}) \cdot m_{\lambda_{t+1}, z_{t+1}}(z_{t+1})) \cdot \exp\{\langle \omega_{t+1}, t(z_t, z_{t+1})\rangle\} dz_{t+1} \tag{102}$$

$$\propto m_{z_t, \omega_{t+1}}(z_t) \cdot \int \frac{q(z_{t+1})}{m_{\omega_{t+1}, z_{t+1}}(z_{t+1})} \cdot \exp\{\langle \omega_{t+1}, t(z_t, z_{t+1})\rangle\} dz_{t+1} \tag{103}$$

Now let's evaluate the integral:

$$\int \exp\left\{\left\langle \begin{bmatrix} -\frac{1}{2}J_{11,t} \\ J_{12,t} \\ -\frac{1}{2}J_{22,t} \\ -h_{1,t} \\ h_{2,t} \end{bmatrix}, \begin{bmatrix} z_t z_t^T \\ z_t z_{t+1}^T \\ z_{t+1} z_{t+1}^T \\ z_t \\ z_{t+1} \end{bmatrix} \right\rangle + \left\langle \begin{bmatrix} f_{t+1|T} - f_{t+1|t} \\ -\frac{1}{2}(F_{t+1|T} - F_{t+1|t}) \end{bmatrix}, \begin{bmatrix} z_{t+1} \\ z_{t+1} z_{t+1}^T \end{bmatrix} \right\rangle \right\} dz_{t+1} \quad (104)$$

$$= \exp\left\{\left\langle \begin{bmatrix} -\frac{1}{2}J_{11,t} \\ -h_{1,t} \end{bmatrix}, \begin{bmatrix} z_t z_t^T \\ z_t \end{bmatrix} \right\rangle \right\} \int_{z_{t+1}} \exp\left\{\left\langle \begin{bmatrix} f_{t+1|T} - f_{t+1|t} + h_{2,t} + J_{12,t}^T z_t \\ -\frac{1}{2}(F_{t+1|T} - F_{t+1|t} + J_{22,t}) \end{bmatrix}, \begin{bmatrix} z_{t+1} \\ z_{t+1} z_{t+1}^T \end{bmatrix} \right\rangle \right\} dz_{t+1}$$
$$\quad (105)$$

$$= \exp\left\{\left\langle \begin{bmatrix} -\frac{1}{2}J_{11,t} \\ -h_{1,t} \end{bmatrix}, \begin{bmatrix} z_t z_t^T \\ z_t \end{bmatrix} \right\rangle + \log Z(f_{t+1|T} - f_{t+1|t} + h_{2,t} + J_{12,t}^T z_t, F_{t+1|T} - F_{t+1|t} + J_{22,t})\right\}$$
$$\quad (106)$$

Again, let's evaluate the log partition function at the end, simplifying $C_t = F_{t+1|T} - F_{t+1|t} + J_{22,t}$:

$$\frac{1}{2}(f_{t+1|T} - f_{t+1|t} + h_{2,t} + J_{12,t}^T z_t)^T C_t^{-1}(f_{t+1|T} - f_{t+1|t} + h_{2,t} + J_{12,t}^T z_t) - \frac{1}{2}\log|C_t| \quad (107)$$

We recover linear term

$$z_t^T J_{12,t} C_t^{-1}(f_{t+1|T} - f_{t+1|t} + h_{2,t}) \quad (108)$$

And quadratic term

$$\frac{1}{2} z_t^T J_{12,t} C_t^{-1} J_{12,t}^T z_t \quad (109)$$

Combining these with the $J_{11,t}$ and $h_{1,t}$ in Eq. (106) and the filtered distributions outside the integral in Eq. (103), we arrive at the update equations above.

**Derivation of Sampler:** Moving backwards, we want to sample from

$$q(z_t | z_{t+1}) \propto m_{z_t, \omega_{t+1}}(z_t) \cdot \exp\{\langle \omega_{t+1}, t(z_t, z_{t+1})\rangle\} \quad (110)$$

The rest of the derivation is easy.

## D.2 Parallel LDS Inference

This adaptation of the Kalman smoother algorithm is highly sequential and therefore inefficient on GPU architectures. We adapt the technique of Särkkä & García-Fernández [56] to parallelize our algorithm across time steps.

Let us define the *kernel* of a (possibly un-normalized) exponential family distribution as the inner product portion of the distribution:

$$\ker(\exp\{\langle \eta, t(z)\rangle - c\}) = \exp\{\langle \eta, t(z)\rangle\} \quad (111)$$

Note that $\ker(f(z)) \propto f(z)$. We define two sequences of functions:

$$g_t(z_{t-1}) = \ker\left(\int_{z_t} \exp\{\langle t(\theta_t), t(z_{t-1}, z_t)\rangle\} \cdot \exp\{\langle \lambda_t, t(z_t)\rangle\} dz_t\right) \quad (112)$$

$$= \exp\left\{\left\langle \begin{bmatrix} -\frac{1}{2}(J_{11,t} - J_{12,t}(R_t + J_{22,t})^{-1}J_{12,t}^T) \\ -h_{1,t} + J_{12,t}(R_t + J_{22,t})^{-1}(r_t + h_2) \end{bmatrix}, \begin{bmatrix} z_{t-1} z_{t-1}^T \\ z_{t-1} \end{bmatrix} \right\rangle \right\} \quad (113)$$

$$= \exp\left\{\left\langle \begin{bmatrix} -\frac{1}{2}\Gamma_t \\ \gamma_t \end{bmatrix}, \begin{bmatrix} z_{t-1} z_{t-1}^T \\ z_{t-1} \end{bmatrix} \right\rangle \right\} \quad (114)$$

Thus $g_t(z_{t-1})$ is defined by natural parameters $\Gamma_t, \gamma_t$. For $t = 1$, we set $\Gamma_1 = 0$, $\gamma_1 = 0$ as there is no time-step 0. Thus we claim $g_1(z_0) = 1$ is a uniform function.

Further define $f_t(z_{t-1}, z_t)$

$$f_t(z_{t-1}, z_t) = \exp\{\langle t(\theta_t), t(z_{t-1}, z_t)\rangle\} \cdot \exp\{\langle \lambda_t, t(z_t)\rangle\}/g_t(z_{t-1}) \tag{115}$$

$$= \exp\left\{\left\langle \begin{bmatrix} -\frac{1}{2}(J_{11,t} - \Gamma_t) \\ -(h_{1,t} - \gamma_t) \\ J_{12,t} \\ h_{2,t} + r_t \\ -\frac{1}{2}(J_{22,t} + R_t) \end{bmatrix}, \begin{bmatrix} z_{t-1}z_{t-1}^T \\ z_{t-1} \\ z_{t-1}z_t^T \\ z_t \\ z_t z_t^T \end{bmatrix} \right\rangle\right\} \tag{116}$$

$$= \exp\left\{\left\langle \begin{bmatrix} -\frac{1}{2}\Phi_{11,t} \\ -\phi_{1,t} \\ \Phi_{12,t} \\ \phi_{2,t} \\ -\frac{1}{2}\Phi_{22,t} \end{bmatrix}, \begin{bmatrix} z_{t-1}z_{t-1}^T \\ z_{t-1} \\ z_{t-1}z_t^T \\ z_t \\ z_t z_t^T \end{bmatrix} \right\rangle\right\} \tag{117}$$

We similarly define $f_1(z_0, z_1) = f_1(z_1)$, or equivalently $\Phi_{11,1} = \Phi_{12,1} = 0$ and $\phi_{1,1} = 0$ as there is no $z_0$. $\phi_{2,t} = h_0 + r_1$ and $\Phi_{22,t} = J_0 + R_1$ to include the factors $h_0, J_0$ from the initial state distribution $p(z_1)$.

We define the associative operation:

$$(f_i, g_i) \otimes (f_j, g_j) = (f_{ij}, g_{ij}) \tag{118}$$

Where

$$f_{ij}(a, c) = \ker\left(\frac{\int g_j(b)f_j(b,c)f_i(a,b)db}{\int g_j(b)f_i(a,b)db}\right) \tag{119}$$

And

$$g_{ij}(a) = \ker\left(g_i(a)\int g_j(b)f_i(a,b)db\right) \tag{120}$$

**Proposition D.1.** *The operation $\otimes$ is associative.*

*Proof.* Särkkä & García-Fernández [56] define an operator

$$(f_i, g_i) \times (f_j, g_j) = (\hat{f}_{ij}, \hat{g}_{ij}) \tag{121}$$

Where

$$f_{ij}(a, c) = \frac{\int g_j(b)f_j(b,c)f_i(a,b)db}{\int g_j(b)f_i(a,b)db} \tag{122}$$

And

$$g_{ij}(a) = g_i(a)\int g_j(b)f_i(b|a)db \tag{123}$$

and prove that this operator is associative. Further, any rescaling of inputs leads to a simple rescaling of outputs due to the multi-linear nature of these functions. Thus for constants $c_1, c_2, c_3, c_4$ there exist constants $c_5, c_6$ such that

$$(c_1 f_i, c_2 g_i) \times (c_3 f_j, c_4 g_j) = (c_5 \hat{f}_{ij}, c_6 \hat{g}_{ij}) \tag{124}$$

This gives us that

$$((f_i, g_i) \otimes (f_j, g_j)) \otimes (f_k, g_k) = (\ker(\hat{f}_{ij}), \ker(\hat{g}_{ij})) \otimes (f_k, g_k) \tag{125}$$

$$= (c_1 \hat{f}_{ij}, c_2 \hat{g}_{ij}) \otimes (f_k, g_k) \tag{126}$$

$$= \ker\left((c_1 \hat{f}_{ij}, c_2 \hat{g}_{ij}) \times (f_k, g_k)\right) \tag{127}$$

$$= \ker\left((\hat{f}_{ij}, \hat{g}_{ij}) \times (f_k, g_k)\right) \tag{128}$$

A similar decomposition shows that

$$(f_i, g_i) \otimes ((f_j, g_j) \otimes (f_k, g_k)) = \ker\left((f_i, g_i) \times (\hat{f}_{jk}, \hat{g}_{jk})\right) \tag{129}$$

$$= \ker\left((\hat{f}_{ij}, \hat{g}_{ij}) \times (f_k, g_k)\right) \tag{130}$$

$$= ((f_i, g_i) \otimes ((f_j, g_j)) \otimes (f_k, g_k) \tag{131}$$

Where the second equality comes from the associativity of the $\times$ operator. $\qquad\square$

Finally, we need to show that this operation gives us the right results:

**Proposition D.2.** *Let $a_i = (f_i, g_i)$. Then*

$$a_1 \otimes a_2 \otimes \cdots \otimes a_t = (f_{1:t}, g_{1:t}) \tag{132}$$

*where*

$$f_{1:t}(z_t) = \exp\left\{ \left\langle \begin{bmatrix} f_{t|t} \\ -\frac{1}{2}F_{t|t} \end{bmatrix}, \begin{bmatrix} z_t \\ z_t z_t^T \end{bmatrix} \right\rangle \right\} \tag{133}$$

*is parameterized by the filtered messages $f_{t|t}, F_{t|t}$.*

*Proof.* We prove by induction that this holds when the operations are performed sequentially, namely

$$(((((a_1 \otimes a_2) \otimes a_3) \otimes \dots) \otimes a_t \tag{134}$$

These "left-justified" operations are simpler, and thus allow us to easily demonstrate correctness. We then can make use of the associative property of $\otimes$ to show that no matter the order of operations we get the correct result.

Recall that $g_1(z_0) = 1$ is the uniform function (as there is no $z_0$) and thus has $\Gamma_1 = 0, \gamma_1 = 0$. By induction, this must be true of *all* the cumulative kernels $g_{ij}(z_0)$, as if $g_i(a)$ in Eq. (120) is independent of $a$, then $g_{ij}(a)$ must be, too. Similarly, we recall that $f_1(z_0, z_1) = f_1(z_1)$ is independent of its first argument and thus has $\Phi_{11,1} = \Phi_{12,1} = 0$ and $\phi_{1,1} = 0$. Again, by induction *all* of the cumulative kernels $f_{1:t}(z_0, z_t)$ must have their first three parameters equal to 0; in Eq. (119) if $f_i(a, b)$ is independent of $a$ then $f_{ij}(a, c)$ will be, too. Thus $f_{1:t}(z_0, z_t) = f_{1:t}(z_t)$.

Next we show that $f_{1:t}(z_t) = m_{z_t,\omega}(z_t)$, the filtered messages from the sequential filter. For $t = 1$, $f_{1:1}(z_1) = m_{z_1,\omega}(z_1)$ by construction. We then see:

$$f_{1:t+1}(z_{t+1}) = \ker\left( \frac{\int_{z_t} g_{t+1}(z_t) f_{t+1}(z_{t+1}) m_{z_t,\omega}(z_t) dz_t}{\int g_{t+1}(z_t) f_{1:t}(z_t) dz_t} \right) \tag{135}$$

$$= \ker\left( \int_{z_t} g_{t+1}(z_t) f_{t+1}(z_{t+1}) m_{z_t,\omega}(z_t) dz_t \right) \tag{136}$$

$$= \ker\left( \int_{z_t} \exp\{\langle \omega_{t+1}, t(z_t, z_{t+1}) \rangle\} \cdot \exp\{\langle \lambda_{t+1}, t(z_{t+1}) \rangle\} \cdot m_{z_t,\omega}(z_t) dz_t \right) \tag{137}$$

$$= \ker\left( \exp\{\langle \lambda_{t+1}, t(z_{t+1}) \rangle\} \cdot \int_{z_t} \exp\{\langle t(\theta_{t+1}), t(z_t, z_{t+1}) \rangle\} \cdot m_{z_t,\omega}(z_t) dz_t \right) \tag{138}$$

$$= \ker\left( \exp\{\langle \lambda_{t+1}, t(z_{t+1}) \rangle\} \cdot m_{z_{t+1},\omega}(z_{t+1}) \right) = m_{z_{t+1},\omega_{t+1}}(z_{t+1}) \tag{139}$$

Completing our proof! $\qquad\square$

This algorithm produces correct filtered messages but due to its associativity can be computed in parallel. To complete the filtering algorithm, we outline how to evaluate $(f_i, f_j) \otimes (g_i, g_j)$. The integrals in Eqs. (119), (120) can be evaluated as simple functions of the parameters of $f_i, f_j, g_i, g_j$:

**Proposition D.3.** *Let*

$$f_i(a, b) = \exp\left\{ \left\langle \begin{bmatrix} -\frac{1}{2}\Phi_{11,i} \\ -\phi_{1,i} \\ \Phi_{12,i} \\ \phi_{2,i} \\ -\frac{1}{2}\Phi_{22,i} \end{bmatrix}, \begin{bmatrix} aa^T \\ a \\ ab^T \\ b \\ bb^T \end{bmatrix} \right\rangle \right\}, \qquad f_j(b, c) = \exp\left\{ \left\langle \begin{bmatrix} -\frac{1}{2}\Phi_{11,j} \\ -\phi_{1,j} \\ \Phi_{12,j} \\ \phi_{2,j} \\ -\frac{1}{2}\Phi_{22,j} \end{bmatrix}, \begin{bmatrix} bb^T \\ b \\ bc^T \\ c \\ cc^T \end{bmatrix} \right\rangle \right\} \tag{140}$$

*and*

$$g_i(a) = \exp\left\{ \left\langle \begin{bmatrix} -\frac{1}{2}\Gamma_i \\ \gamma_i \end{bmatrix}, \begin{bmatrix} aa^T \\ a \end{bmatrix} \right\rangle \right\}, \qquad g_j(b) = \exp\left\{ \left\langle \begin{bmatrix} -\frac{1}{2}\Gamma_j \\ \gamma_j \end{bmatrix}, \begin{bmatrix} bb^T \\ b \end{bmatrix} \right\rangle \right\} \tag{141}$$

*Then given the following definitions*

$$C_{ij} = \Gamma_j + \Phi_{22,i} \tag{142}$$

$$\Gamma_{ij} = (\Phi_{11,i} - \Phi_{12,i} C_{ij}^{-1} \Phi_{12,i}^T) \tag{143}$$

$$\gamma_{ij} = -\phi_{1,i} + \Phi_{12,i} C_{ij}^{-1} (\gamma_j + \phi_{2,i}) \tag{144}$$

$$P_{ij} = \Gamma_j + \Phi_{22,i} + \Phi_{11,i} \tag{145}$$

*We can compute* $(f_{ij}, g_{ij}) = (f_i, g_i) \otimes (f_j, g_j)$ *as:*

$$f_{ij}(a,c) = \exp\left\{ \left\langle \begin{bmatrix} -\frac{1}{2}(\Phi_{11,i} - \Phi_{12,i} P_{ij}^{-1} \Phi_{12,i}^T - \Gamma_{ij}) \\ -(\phi_{1,i} - \Phi_{12,i} P_{ij}^{-1}(\phi_{2,i} - \phi_{1,j} + \gamma_j) + \gamma_{ij}) \\ \Phi_{12,i} P_{ij}^{-1} \Phi_{12,j} \\ \phi_{2,j} + \Phi_{12,j}^T P_{ij}^{-1}(\phi_{2,i} - \phi_{1,j} + \gamma_j) \\ -\frac{1}{2}(\Phi_{22,j} - \Phi_{12,j}^T P_{ij}^{-1} \Phi_{12,j}) \end{bmatrix}, \begin{bmatrix} aa^T \\ a \\ ac^T \\ c \\ cc^T \end{bmatrix} \right\rangle \right\} \tag{146}$$

*and*

$$g_{ij}(c) = \exp\left\{ \left\langle \begin{bmatrix} -\frac{1}{2}(\Gamma_i + \Gamma_{ij}) \\ \gamma_i + \gamma_{ij} \end{bmatrix}, \begin{bmatrix} cc^T \\ c \end{bmatrix} \right\rangle \right\} \tag{147}$$

The proof of this involves integrating the kernels of Gaussians, which is done many times in this supplement; it is left as an exercise for the reader.

The last task is to define the parallel smoother. The construction is very similar to the filter. Let

$$e_t(z_t, z_{t+1}) = \frac{m_{z_t, \omega_{t+1}}(z_t)}{m_{\omega_{t+1}, z_{t+1}}(z_{t+1})} \cdot \exp\{\langle \omega_{t+1}, t(z_t, z_{t+1}) \rangle\} \tag{148}$$

$$= \exp\left\{ \left\langle \begin{bmatrix} -\frac{1}{2}(J_{11,t} + F_{t|t}) \\ -(h_{1,t} - f_{t|t}) \\ J_{12,t} \\ h_{2,t} + r_{t+1} - h_{t+1|t+1} \\ -\frac{1}{2}(J_{22,t} + R_{t+1} - J_{t+1|t+1}) \end{bmatrix}, \begin{bmatrix} z_t z_t^T \\ z_t \\ z_t z_{t+1}^T \\ z_{t+1} \\ z_{t+1} z_{t+1}^T \end{bmatrix} \right\rangle \right\} \tag{149}$$

$$= \exp\left\{ \left\langle \begin{bmatrix} -\frac{1}{2} E_{11,t} \\ -\epsilon_{1,t} \\ E_{12,t} \\ \epsilon_{2,t} \\ -\frac{1}{2} E_{22,t} \end{bmatrix}, \begin{bmatrix} z_t z_t^T \\ z_t \\ z_t z_{t+1}^T \\ z_{t+1} \\ z_{t+1} z_{t+1}^T \end{bmatrix} \right\rangle \right\} \tag{150}$$

Where we let $e_T(z_T, z_{T+1}) = e_T(z_T)$ with $E_{11,T} = J_{T|T}, \epsilon_{1,T} = -h_{T|T}$ and remaining parameters equal to 0 (as there is no $z_{T+1}$. Then the associative operator $\oplus$ is given by

$$e_{ij}(a,c) = e_i(a,b) \oplus e_j(b,c) = \ker\left( \int_b e_i(a,b) e_j(b,c) db \right) \tag{151}$$

Which can be computed (derivations omitted) as

$$D_{ij} = E_{22,i} + E_{11,j} \tag{152}$$

$$e_{ij}(a,c) = \exp\left\{ \left\langle \begin{bmatrix} -\frac{1}{2}(E_{11,i} - E_{12,i} D_{ij}^{-1} E_{12,i}^T) \\ -(\epsilon_{1,i} - E_{12,i} D_{ij}^{-1}(\epsilon_{2,i} - \epsilon_{1,j})) \\ E_{12,i} D_{ij}^{-1} E_{12,j} \\ \epsilon_{2,j} + E_{12,j}^T D_{ij}^{-1}(\epsilon_{2,i} - \epsilon_{1,j}) \\ -\frac{1}{2}(E_{22,j} - E_{12,j}^T D_{ij}^{-1} E_{12,j}) \end{bmatrix}, \begin{bmatrix} aa^T \\ a \\ ac^T \\ c \\ cc^T \end{bmatrix} \right\rangle \right\} \tag{153}$$

Särkkä & García-Fernández [56] prove this operation is associative when operating on full distributions (i.e. without taking the kernel). The proof that we retain associativity when removing multiplicative constants mirrors the proof of Prop. D.1 and is omitted for brevity. Finally, we show that when these operations are done sequentially,

$$e_t \oplus (e_{t+1} \oplus (e_{t+1} \oplus \ldots (e_{T-1} \oplus e_T))))) = q(z_t) \tag{154}$$

The base $e_T(z_t) = q(z_T)$ is true by construction. Then, plugging the inductive hypothesis and Eq. (148) into the definition of $\oplus$ yields Eq. (103), proving that $e_t(z_t) = q(z_t)$ implies $e_{t-1}(z_{t-1}) = q(z_{t-1})$.

### D.3 HMM Inference

The discrete Markov model is time homogenous and requires only expected probabilities for the initial state $k_2$ and expected probabilities for the transition matrix $\pi_{ij}$. The "observations" of the HMM are (in the case of the SLDS) messages passed from the LDS chain, which we denote $\omega_t(k)$. The setup is fundamentally the same as the LDS:

**Input:** $\mathbb{E}_{q(\theta)}[t(\theta)]$, $\eta_{\text{recog}}$.

**Forward Pass:**

$$\alpha_1(k) = \mathbb{E}[\log \pi_k] + \omega_1(k) \tag{155}$$

$$\texttt{for } t = 2, \ldots, T \quad \alpha_t(k) = \omega_t(k) + \texttt{logsumexp}_{i=1}^K \big[ \mathbb{E}[\log \pi_{ik}] + \alpha_{t-1}(i) \big] \tag{156}$$

$$\log Z = \texttt{logsumexp}_{k=1}^K \alpha_T(k) \tag{157}$$

**Backward Pass:**

$$\beta_T(k) = 0 \tag{158}$$

$$\texttt{for } t = T-1, \ldots, 1 \quad \beta_t(k) = \texttt{logsumexp}_{j=1}^K \big[ \omega_{t+1}(j) + \mathbb{E}[\log \pi_{kj}] + \beta_{t+1}(j) \big] \tag{159}$$

**Return:** log marginals $\alpha_t(k) + \log \beta_t(k)$; $\log Z$

## E   Evaluating the SVAE loss

Johnson et al. [29] decompose the loss into three terms

$$-\mathcal{L}[q(\theta)q(z), \gamma] = \overbrace{\text{KL}(q(\theta) \parallel p(\theta)))}^{\text{Prior KL}} + \overbrace{\mathbb{E}_{q(\theta)} \text{KL}(q(z) \parallel p(z|\theta))}^{\text{Local KL}} - \overbrace{\mathbb{E}_{q(z)}[p_\gamma(x|z)]}^{\text{Reconstruction}} \tag{160}$$

Because $p(\theta), p(z|\theta), q(\theta; \eta), q_\phi(z|x; \eta)$ are all exponential family distributions, the expectations here can be evaluated in closed form, with the exception of the reconstruction loss $\mathbb{E}_{q(z)}[p_\gamma(x|z)]$. The reconstruction loss can be estimated without bias using Monte Carlo samples and the reparameterization trick [32]. For this to work, we must be able to reparameterize the sample of $q(z)$ such that we can back propagate through the sampling routine, which is not possible for discrete random variables. As a result, the latent variables which need to be sampled to reconstruct $x$ (i.e. all variables which are in the same mean field component as a variable being fed into the encoder) must be continuous.

The first term of the loss, the prior KL, is a regularizer on the graphical model parameters and can be easily evaluated via Eq. (25). The rest of this section is devoted to computing the KL divergence between the prior and variational factor of the local latent varibles $z$. We will start with the fully structured case where no mean field separation occurs.

Recalling that the optimal $q(z)$ has parameters $\omega$ given by Eq. (47), we can again invoke Eq. (25) to get

$$\mathbb{E}_{q(\theta)} \text{KL}(q(z) \parallel p(z|\theta)) = \mathbb{E}_{q(\theta)}[\langle \omega - t(\theta), E_{q(z)} t(z) \rangle - \log Z(\omega) + \log Z(t(\theta))] \tag{161}$$

$$= \langle \omega - \mathbb{E}_{q(\theta)}[t(\theta)], E_{q(z)} t(z) \rangle - \log Z(\omega) + \log \mathbb{E}_{q(\theta)}[Z(t(\theta))] \tag{162}$$

$$= \langle \lambda_\phi(x), E_{q(z)} t(z) \rangle - \log Z(\omega) + \log \mathbb{E}_{q(\theta)}[Z(t(\theta))] \tag{163}$$

Namely, because $\log q(z)$ is defined as the expected log prior $p(z|\theta)$ plus the recognition potentials, these two distributions only differ in log space by those recognition potentials. The log partition function of the graphical model can be obtained by belief propagation, and the expected log partition function from the prior can be generally known in closed form as the sum of individual partition functions at each vertex or factor in the corresponding graphical model.

**Example: LDS**   The LDS SVAE only outputs recognition potentials on a subset of the statics of $q(z)$. In particular, the encoder takes the observations $x$ and outputs normal likelihood functions on each $z$ independently. Thus the cross-temporal covariance statistics $z_t z_{t+1}^T$ have no corresponding recognition potentials. Further, we restrict the potentials on $z_t z_t^T$ to be diagonal, meaning our observational uncertainty factorizes across dimensions. Thus the only statistics in $t(z)$ for which $\lambda_\phi(x)$ are non-zero are $z_t$ and $z_{ti}^2$ for each component $i$ of $t(z)$. As a result, the inner product $\langle \lambda_\phi(x), \mathbb{E}_{q(z)} t(z) \rangle$ sums over $2TD$ terms for a sequence with $T$ timesteps and $D$-dimensional latent variable at each step.

**Local KL with structured mean field** When there is mean field separation, the prior and variational factor differ in more than just the recognition potentials, as $q(z)$ replaces some factors of $p(z|\theta)$ with separable factors. We can rewrite

$$\mathrm{KL}(q(z) \parallel p(z|\theta)) = \sum_{m=1}^{M} \mathrm{KL}(q(z_m) \parallel p(z_m|z_{<m}, \theta)) \tag{164}$$

The natural parameters of $q(z_m)$ differ from the natural parameters of $p(z_m|z_{<m}, \theta)$ not only in the recognition potentials, but also for any factors which include a variable in "later" clusters $z_{>m}$.

**Example: SLDS** The SLDS SVAE has 2 mean field clusters, $q(k)q(z)$ for the discrete variables $k$ and continuous variables $z$. If we write our prior distribution $p(\theta)p(k|\theta)p(z|k, \theta)$ then $\mathrm{KL}(q(z) \parallel p(z|k, \theta))$ is computed exactly the same as in the LDS example above, as these two distributions differ only in the recognition potentials. The new challenge is computing $\mathrm{KL}(q(k) \parallel p(k|\theta))$ which contains no recognition potentials but differs based on mean field messages from $q(z)$ which are present in $q(k)$ but not in $p(k|\theta)$.

In particular, the prior $p(k|\theta)$ is a time series with no observations, while $q(k)$ includes parameters at every time step which encode the log probability of being in that state at that time, given by Eq. (60): $\mathbb{E}_{q(\theta)}[t(\theta_{\mathrm{trans}})] \cdot \mathbb{E}_{q(z)}[t(z_{t-1}, z_t)]$. The corresponding sufficient statistics are the marginal probabilities of being in state $k$ at time $t$. Note that we never need to compute the sufficient statistics corresponding to HMM transitions because we don't need them for the local KL nor do we need them to sample from the HMM (which we never do).

**Surrogate Loss** The surrogate loss differs from the true loss only in the reconstruction term, so it can similarly be written

$$-\hat{\mathcal{L}}[q(\theta)q(z), \phi] = \overbrace{\mathrm{KL}(q(\theta) \parallel p(\theta))}^{\text{Prior KL}} + \overbrace{\mathbb{E}_{q(\theta)}\mathrm{KL}(q(z) \parallel p(z|\theta))}^{\text{Local KL}} - \langle \lambda_\phi(x), \mathbb{E}_{q(z)}t(z) \rangle \tag{165}$$

This is trivial to evaluate, as the local KL contains the negation of the last term (see Eq. (163) for the structured case. The mean field case contains these terms as well in addition to ones corresponding to "pulled apart" factors) and thus the surrogate loss can be computed by omitting the recognition-potential-times-expected-statistics part of the local KL.

# F   Natural gradients

Here we dissect two claims in the main paper more closely. First, we will show how natural gradients with respect to $\eta$ can be easily computed. Defining, as in the main paper, $\mu_\eta = \mathbb{E}_{q(\theta;\eta)}[t(\theta)]$, we can use the chain rule to write:

$$\frac{\partial \mathcal{L}}{\partial \eta} = \frac{\partial \mathcal{L}}{\partial \mu_\eta}\frac{\partial \mu_\eta}{\partial \eta} + \frac{\partial \mathcal{L}}{\partial \eta}_{\,\mathtt{detach}(\mu_\eta)} \tag{166}$$

Where $\frac{\partial \mathcal{L}}{\partial \eta}_{\,\mathtt{detach}(\mu_\eta)}$ refers to the gradient of the loss with respect to $\eta$, disconnecting the computational graph at $\mu_\eta$ (i.e. the dependence of the loss on $\eta$ which is not accounted for by its dependence on $\mu_\eta$).

**Proposition F.1.**

$$\frac{\partial \mathcal{L}}{\partial \eta}_{\,\mathit{detach}(\mu_\eta)} = 0 \tag{167}$$

*Proof.* $q(z)$ depends on $\eta$ only through the expected statistics $\mu_\eta$ which define the parameters of the graphical model factors. Further, the local KL, outlined in Eq. (163), only depends on $q(\theta)$ through its expected statistics (noting that the log normalizers of $p(z|\theta)$ are also staistics of $q(\theta)$). Thus the only place in the loss that $\eta$ appears on its own is the prior KL. Let $\eta_0$ be the natural parameters of the prior $p(\theta)$. Then the prior KL, as per Eq. (25), is given by:

$$\mathrm{KL}(q(\theta) \parallel p(\theta)) = \langle \eta - \eta_0, \mu_\eta \rangle - \log Z(\eta) + \log Z(\eta_0) \tag{168}$$

Then we see:

$$\frac{\partial}{\partial \eta} \mathrm{KL}(q(\theta) \parallel p(\theta)) = \frac{\partial \mathrm{KL}(q(\theta) \parallel p(\theta))}{\partial \mu_\eta} \frac{\partial \mu_\eta}{\partial \eta} + \mu_\eta - \frac{\partial}{\partial \eta} \log Z(\eta) \tag{169}$$

$$= \frac{\partial \mathrm{KL}(q(\theta) \parallel p(\theta))}{\partial \mu_\eta} \frac{\partial \mu_\eta}{\partial \eta} = (\eta - \eta_0) \frac{\partial \mu_\eta}{\partial \eta} \tag{170}$$

Where we use Prop. C.1 to cancel terms. Thus we conclude that the entire gradient of the loss with respect to $\eta$ passes through $\mu_\eta$ □

The main consequence of this theorem is that in all cases the gradient can be written

$$\frac{\partial \mathcal{L}}{\partial \eta} = \frac{\partial \mathcal{L}}{\partial \mu_\eta} \frac{\partial \mu_\eta}{\partial \eta} \tag{171}$$

And thus the natural gradient, which multiples the gradient by the inverse of $\frac{\partial \mu_\eta}{\partial \eta}$, can be easily computed withoout matrix arithmetic by skipping the portion of gradient computation corresponding to the $\eta \to \mu_\eta$ mapping. Here is some sample Jax code which takes a function `f` and returns a function which computes `f` in the forward pass but in the backward pass treats the gradient as identity:

```
def straight_through(f):
    def straight_through_f(x):
        zero = x - jax.lax.stop_gradient(x)
        return  zero + jax.lax.stop_gradient(f(x))
    return straight_through_f
```

**Fisher information of transformed variables**   . Given a distribution $q(\theta; \eta)$ with natural parameters $\eta$, the Fisher information matrix of $\eta$ is given by:

$$F_\eta = \mathbb{E}_{q(\theta)} \left[ \left( \frac{\partial \log q(\theta)}{\partial \eta} \right)^T \cdot \left( \frac{\partial \log q(\theta)}{\partial \eta} \right) \right] \tag{172}$$

Note that for an exponential family distribution

$$\frac{\partial \log q(\theta)}{\partial \eta} = t(\theta) - \mathbb{E}_{q(\theta)}[t(\theta)] \tag{173}$$

which clearly has expectation 0 under $q(\theta)$. Thus

$$F_\eta = \mathbb{E}_{q(\theta)} \left[ (t(\theta) - \mathbb{E}_{q(\theta)}[t(\theta)])^T \cdot (t(\theta) - \mathbb{E}_{q(\theta)}[t(\theta)]) \right] \tag{174}$$

$$= \mathrm{Var}_{q(\theta)}[t(\theta)] \tag{175}$$

Exponential family theory gives us further equalities:

$$\mathrm{Var}_{q(\theta)}[t(\theta)] = \frac{\partial^2}{\partial \eta^2} \log Z(\eta) = \frac{\partial}{\partial \eta} \mathbb{E}_{q(\theta)}[t(\theta)] \tag{176}$$

Thus deriving our desired property that $F_\eta = \frac{\partial \mu_\eta}{\partial \eta}$.

Now we consider a more general case, where $\eta = f(\tilde{\eta})$ for some unconstrained, non-natural parameters $\tilde{\eta}$ and continuous invertible function $f$. Then we see:

$$F_{\tilde{\eta}} = \mathbb{E}_{q(\theta)} \left[ \left( \frac{\partial \log q(\theta)}{\partial \tilde{\eta}} \right)^T \cdot \left( \frac{\partial \log q(\theta)}{\partial \tilde{\eta}} \right) \right] \tag{177}$$

$$= \mathbb{E}_{q(\theta)} \left[ \left( \frac{\partial \log q(\theta)}{\partial \eta} \cdot \frac{\partial \eta}{\partial \tilde{\eta}} \right)^T \cdot \left( \frac{\partial \log q(\theta)}{\partial \eta} \cdot \frac{\partial \eta}{\partial \tilde{\eta}} \right) \right] \tag{178}$$

$$= \frac{\partial \eta}{\partial \tilde{\eta}}^T \cdot \mathbb{E}_{q(\theta)} \left[ \left( \frac{\partial \log q(\theta)}{\partial \eta} \right)^T \cdot \left( \frac{\partial \log q(\theta)}{\partial \eta} \right) \right] \cdot \frac{\partial \eta}{\partial \tilde{\eta}} \tag{179}$$

$$= \frac{\partial \eta}{\partial \tilde{\eta}}^T \cdot F_\eta \cdot \frac{\partial \eta}{\partial \tilde{\eta}} \tag{180}$$

We can use this to compute the natural gradient with respect to the unconstrained parameters:

$$\frac{\partial \mathcal{L}}{\partial \tilde{\eta}} F_{\tilde{\eta}}^{-1} = \left( \frac{\partial \mathcal{L}}{\partial \mu} \cdot \frac{\partial \mu}{\partial \eta} \cdot \frac{\partial \eta}{\partial \tilde{\eta}} \right) \cdot \left( \frac{\partial \eta}{\partial \tilde{\eta}}^{-1} \cdot F_\eta^{-1} \cdot \frac{\partial \eta}{\partial \tilde{\eta}}^{-T} \right) = \frac{\partial \mathcal{L}}{\partial \mu} \cdot \frac{\partial \eta}{\partial \tilde{\eta}}^{-T} = \left( \frac{\partial \tilde{\eta}}{\partial \eta} \cdot \nabla_\mu \mathcal{L} \right)^T \tag{181}$$

**Implementing natural gradients via automatic differentiation**  We argued in the main paper that to implement natural gradients we replace the reverse-mode backpropagation through the $\tilde{\eta} \to \eta$ map with a forward-mode differentiation through the inverse operation. Assuming we have a forward map `f` and inverse function `f_inv`, sample code is shown below:

```
f_natgrad = jax.custom_vjp(f)

def f_natgrad_fwd(input):
    return f(input), f(input)

def f_natgrad_bwd(resids, grads):
    return (jax.jvp(f_inv, (resids,), (grads,))[1],)

f_natgrad.defvjp(f_natgrad_fwd, f_natgrad_bwd)
```

## G  Matrix-normal exponential family distributions

Recall that exponential family distributions are given by

$$p(x; \eta) = \exp\{\langle \eta, t(x) \rangle - \log Z(\eta)\} \tag{182}$$

The natural parameters often have coupled constraints, and so can be expressed as a simple function of *canonical parameters* which carry easily-interpretable information about the distribution (e.g. the natural parameters of a normal distribution can be expressed in terms of the mean and precision matrix of that distribution). Thus to define each distribution, we provide (a) the natural parameters $\eta$ as a function of canonical parameters, (b) the sufficient statistics $t(x)$, (c) the expected sufficient statistics $\mathbb{E}_{p(x;\eta)}[t(x)]$, and (d) the log partition function $\log Z(\eta)$.

Note that sometimes we have symmetric positive definite (SPD) matrix-valued parameters. In every case, the corresponding statistics for those parameters are also symmetric. For example, given a $n \times n$ SPD matrix, a distribution might have parameters and statistics:

$$\eta = \begin{bmatrix} S_{ii} | i = 1, \ldots, n \\ 2 \cdot S_{ij} | i < j \end{bmatrix} \qquad t(x) = \begin{bmatrix} x_i^2 | i = 1, \ldots, n \\ x_i x_j | i < j \end{bmatrix} \tag{183}$$

We note that $\langle \eta, t(x) \rangle = \sum_{i=1}^{n} \sum_{j=1}^{n} S_{ij} x_i x_j$ or equivalently the sum of elements of $S \otimes xx^T$ (for column vector $x$ containing each $x_i$). In the following sections, we will abuse notation for compactness by simply writing:

$$\eta = [S] \qquad t(x) = \begin{bmatrix} xx^T \end{bmatrix} \tag{184}$$

**Constraints.**  To optimize these parameters via gradient descent, we must define constrained parameters as a function of unconstrained parameters as discussed in Sec.[natgrads]. Further, to implement natural gradients we must implement the inverse operation. The table below lists all such transformations.

| Notation | Size | Constraint | $\tilde{\eta} \to \eta$ |
|---|---|---|---|
| $\mathbb{R}^n$ | $n$-length vector | n/a | n/a |
| $\mathbb{R}^{n \times m}$ | $n \times m$ matrix | n/a | n/a |
| $\mathbb{R}_+^n$ | $n$-length vector | All values positive | $\eta = \texttt{Softplus}(\tilde{\eta})$ |
| $\mathbb{R}_{>m}^n$ | $n$-length vector | All values greater than $m$ | $\eta = \texttt{Softplus}(\tilde{\eta})\texttt{+m}$ |
| $\Delta^{n-1}$ | $n$-length vector | Non-negative, sums to 1 | $\eta = \texttt{Softmax}(\tilde{\eta})$ |
| $S_{++}^n$ | $n \times n$ matrix | symmetric positive definite | See below |

Where Softmax is made invertible by appending a 0 to the end of $\tilde{\eta}$. For the SPD matrix parameters, we define $S = \texttt{diag}(\sigma) \cdot R \cdot \texttt{diag}(\sigma)$ where $\sigma \in \mathbb{R}_+^n$ is a vector of standard deviations and $R$ is a correlation matrix, transformed to and from unconstrained space by the tensorflow Correlation Cholesky bijector.

### G.1  Multivariate Normal distribution

If $x \sim \mathcal{N}(\mu, \Sigma)$ for (i) $x, \mu \in \mathbb{R}^n$, and (ii) $\Sigma \in S_{++}^n$, then $p(x; \mu, \Sigma)$ is given by:

$$\eta = \begin{bmatrix} \Sigma^{-1}\mu \\ -\frac{1}{2}\Sigma^{-1} \end{bmatrix} \qquad t(x) = \begin{bmatrix} x \\ xx^T \end{bmatrix} \qquad \mathbb{E}_{p(x;\mu,\Sigma)}[t(x)] = \begin{bmatrix} \mu \\ \mu\mu^T + \Sigma \end{bmatrix} \qquad (185)$$

The log partition function is given by:

$$\log Z(\eta) = \frac{1}{2}\mu^T \Sigma^{-1}\mu + \frac{1}{2}\log|\Sigma| + \frac{n}{2}\log(2\pi) \qquad (186)$$

Here, $\pi \approx 3.14$ is a constant.

### G.2  Normal inverse Wishart (NIW) distribution

If $\mu, \Sigma \sim \mathrm{NIW}(S, m, \lambda, \nu)$ for (i) $\mu, m \in \mathbb{R}^n$, (ii) $\Sigma, S \in S_{++}^n$, (iii) $\lambda \in \mathbb{R}_+$, and (iv) $\nu \in \mathbb{R}_{>n-1}$, then $p(\mu, \Sigma; S, m, \lambda, \nu)$ is given by:

$$\eta = \begin{bmatrix} S + \lambda mm^T \\ \lambda m \\ \lambda \\ \nu + n + 2 \end{bmatrix} \qquad t(\mu, \Sigma) = \begin{bmatrix} -\frac{1}{2}\Sigma^{-1} \\ \Sigma^{-1}\mu \\ -\frac{1}{2}\mu^T\Sigma^{-1}\mu \\ -\frac{1}{2}\log|\Sigma| \end{bmatrix} \qquad \mathbb{E}[t(\mu, \Sigma)] = \begin{bmatrix} -\frac{1}{2}\nu S^{-1} \\ \nu S^{-1}m \\ -\frac{1}{2}(n/\lambda + \nu m^T S^{-1}m) \\ \frac{1}{2}(n\log 2 - \log|S| + \Sigma_{i=0}^{n-1}\psi(\frac{\nu-i}{2})) \end{bmatrix}$$
$$(187)$$

Where $\psi$ is the digamma function. The log partition function is given by:

$$\log Z(\eta) = \frac{\nu}{2} \cdot (n\log 2 - \log|S|) + \log\Gamma_n\left(\frac{\nu}{2}\right) + \frac{n}{2}(\log(2\pi) - \log\lambda) \qquad (188)$$

With the multivariate gamma function $\Gamma_n$.

### G.3  Matrix normal inverse Wishart (MNIW) distribution

If $X, \Sigma \sim \mathrm{MNIW}(S, M, V, \nu)$ for (i) $X, M \in \mathbb{R}^{n \times m}$, (ii) $\Sigma, S \in S_{++}^n$, (iii) $V \in S_{++}^m$, and (iv) $\nu \in \mathbb{R}_{>n-1}$, then $p(X, \Sigma; S, M, V, \nu)$ is given by:

$$\eta = \begin{bmatrix} S + MVM^T \\ MV \\ V \\ \nu + n + m + 1 \end{bmatrix} \qquad t(X, \Sigma) = \begin{bmatrix} -\frac{1}{2}\Sigma^{-1} \\ \Sigma^{-1}X \\ -\frac{1}{2}X^T\Sigma^{-1}X \\ -\frac{1}{2}\log|\Sigma| \end{bmatrix} \qquad \mathbb{E}[t(X, \Sigma)] = \begin{bmatrix} -\frac{1}{2}\nu S^{-1} \\ \nu S^{-1}M \\ -\frac{1}{2}(nV^{-1} + \nu M^T S^{-1}M) \\ n\log 2 - \log|S| + \Sigma_{i=0}^{n-1}\psi(\frac{\nu-i}{2}) \end{bmatrix}$$
$$(189)$$

Where $\psi$ is the digamma function. The log partition function is given by:

$$\log Z(\eta) = \frac{\nu}{2} \cdot (n\log 2 - \log|S|) + \log\Gamma_n\left(\frac{\nu}{2}\right) - \frac{n}{2}|V| + \frac{n \cdot m}{2}\log(2\pi) \qquad (190)$$

With the multivariate gamma function $\Gamma_n$. For the SLDS, we use this MNIW distribution by defining $X \in \mathbb{R}^{n \times n+1} = [A|b]$ where $z_{t+1} \sim \mathcal{N}(Az_t + b, \Sigma)$ so the MNIW distribution provides a prior on the transition matrix, offset, and conditional noise.

