# OpenReview forum: "Unbiased learning of deep generative models with structured discrete representations"
_NeurIPS.cc/2023/Conference — NeurIPS 2023 poster_

### Official Review · Reviewer_gyLw · 2023-07-04

**Soundness:** 2 fair
**Presentation:** 2 fair
**Contribution:** 2 fair
**Rating:** 4
**Confidence:** 4

**Summary:**

The authors revisit learning algorithms for structured variational autoencoders.

This paper proposes using the implicit function theorem to compute the implicit gradients, improving stability and memory efficiency. The authors further revisit the natural gradients studied in the original SVAE paper and propose an unbiased version.

The experimentation is carried out on human pose capture and audio spectrogram datasets. The evaluation is based on modelling performances (likelihood), visual quality (FID score) and missing data imputation.

**Strengths:**

1. This paper is impressive by the depth and breadth of its theory (implicit gradients, unbiased natural gradients)
2. A complete and well-detailed appendix featuring all derivations, although I was not in a position to proof-read all of it
3. The experiment section shows a clear improvement of `SVAE-SLDS` (the method introduced in this paper) over the baseline `SVAE-SLDS-Bias` based on likelihood performances and FID

**Weaknesses:**

1. This paper lacks structure and clarity. More work is needed to synthesize and popularize the main contributions.
2. This paper attempts to make too many contributions and therefore fails to present and support each of them sufficiently.
3. The experiment section is not sufficient (only two rather small datasets).
4. Section 4 claims proposing a more memory efficient algorithm: an experiment demonstrating the applicability to larger-scale datasets is missing.
5. Section 3.2 seems to indicate that discrete latent variable models can only be optimized by reparameterizing discrete distributions. This is wrong, score function estimators (e.g. Reinforce) are viable alternatives.  ["Monte Carlo Gradient Estimation in Machine Learning", Mohamed et al., 2020](https://jmlr.org/papers/volume21/19-346/19-346.pdf) gives a great overview of gradient estimation methods and should, in my opinion, be discussed in the related work section. Reparameterization-free methods exist and work decently. For instance:
	- ["Variational Inference for Monte Carlo Objectives", Mnih et al., 2016](http://proceedings.mlr.press/v48/mnihb16.pdf)
	- ["Revisiting Reweighted Wake-Sleep for Models with Stochastic Control Flow", Anh et al, 2018](https://arxiv.org/abs/1805.10469)
	- ["The Thermodynamic Variational Objective", Masrani et al., 2019](https://arxiv.org/abs/1907.00031)
	- ["Optimal Variance Control of the Score Function Gradient Estimator for Importance Weighted Bounds", Lievin et al., 2020](https://arxiv.org/abs/2008.01998)
	- ["Gradient Estimation with Discrete Stein Operators", Shi et al., 2022](https://arxiv.org/abs/2202.09497)

**Questions:**

1. In the original SVAE paper, the authors explore the case of a simple latent GMM model, why not explore this simple case in this paper?
2. Would your method scale to larger datasets as explored (for instance) in ["Clockwork Variational Autoencoders", Saxena et al., 2021]?
3. Could you please detail how the FID scores are computed?

**Limitations:**

Technical limitations are not clearly discussed.

---

> ### Author Rebuttal · Authors · 2023-08-10
>
> Thank you for your thorough feedback. We will respond to your comments and questions individually.
>
> 1 & 2: Could you provide any specifics for which contributions are not sufficiently supported or not sufficiently connected to the main structure of the paper?
>
> 3: We chose our two datasets to replicate the comparisons made in [1], a recent survey paper of dynamical VAEs. The two datasets consist of 53,443 and 93,215 sequences respectively. This contrasts to related work, which has been restricted to toy and synthetic data: the original SVAE paper [2], the structured inference network paper [3], and recent concurrent work on extending the SVAE (see response to reviewer FFkU) [4] are all only evaluated on synthetic data (except for one experiment in [2] on not-publicly-available data).
>
> 4: Greater memory efficiency does not impact the ability of the model to handle larger datasets, so long as mini-batch learning is used (the relationship between memory efficiency and mini-batch size is demonstrated in Table 1). This improvement does, however, enable easier application of the SVAE to larger models and/or longer sequences which have high memory overhead. Specifically our approach avoids storing *all* of the intermediate values computed within the inner optimization, which would otherwise be need for backpropagation. Even for moderately sized datasets, explicit backpropagation can induce prohibitively high memory costs as shown in table 1.
>
> 5: Thank you for the citations; we will add more discussion of these broader methods to section 3.2. We would like to highlight that, to our knowledge, no one has successfully integrated these methods with the model family we are considering in this paper (dynamical VAEs). As a result, we believe these methods are more suitable for the related work section (as you recommend) than for experimental baselines.
>
> With regard to score function estimators specifically, in preliminary experiments on related models we found these techniques had poor properties. For some published evidence on this front, see [5] which showed that score function estimators have orders of magnitude higher variance than reparameterization-based estimators. [6] compared natural gradient descent to score function estimators and drew the same conclusion.
>
> Q1: The original SVAE paper only explores this simple mixture model on synthetic data. We experimented with this model (and it is implemented in our codebase to be shared), and found that while our methods improve computation and memory efficiency, there was no substantial difference between the models learned by our method and the original SVAE paper. Given this, we felt comparisons on this model were less interesting than the ones included in our submission, but we’d be happy to add an experiment on this model family to our supplement.
>
> Q2: Our paper does not focus on long-range-dependencies (which is its own robust line of work, usually evaluated by temporal prediction tasks rather than the generative model quality/metrics), however our SVAE would be quicker to train on such data than RNN baselines: our derived parallel inference routine has time complexity logarithmic in the sequence length instead of linear.
>
> Q3: Thank you for the question. Details of our FID protocol (and architectures) were missing from our appendix and will be added. The architectures are described in a table (see attached pdf) and are detailed below.
>
> Mocap: The InceptionV3 [7] network typically used to compute Frechet Inception Distance scores was trained on natural images and is therefore inappropriate for evaluating the generative performance of the H3.6M motion capture data. To resolve this, we compute the same score using a different underlying network. We trained a convolutional network to predict which of the 15 different motions was being performed in each sequence (note that in H3.6M, motions are distinct from actions/scenarios, which may include more subtle variations). We used a residual network architecture, where each residual block consists of 2 1-D convolutional layers with batch normalization and a skip connection. We chose 1-D convolutions (across time) as they are more appropriate for our temporal motion capture data. We used the Optuna library to search across model depths, activations and layer sizes to find a network within our design space that performed optimally on a validation dataset (25% of the original training data). As with the original FID score, we use the output of the final average pooling layer to compute the score.
>
> WSJ0: As with the H3.6M dataset, it is necessary to define a different base network to evaluate generative FID scores for the WSJ0 dataset. As the WSJ0 dataset does not have an appropriate set of labels to use to train a classifier, we chose to instead train a classifier on the CREMA-D dataset [9] which consists of similar speech clips. We preprocessed each audio sequence in CREMA-D identically to the WSJ0 dataset, and trained a classifier to predict one of 6 different emotions that was expressed. We used the same setup as H3.6M to choose a network architecture, train the model, and compute FID scores.
>
> [1] See [14] in our paper.
>
> [2] See [19] in our paper.
>
> [3] See [30] in our paper.
>
> [4] Yixiu Zhao, Scott W. Linderman, Revisiting Structured Variational Autoencoders. ICML 2023
>
> [5] Kucukelbir, A., Tran, D., Ranganath, R., Gelman, A., Blei, D., Automatic Differentiation Variational Inference. JMLR 2017
>
> [6] Ji, G., Sujono, D., and Sudderth, E. Marginalized Stochastic Natural Gradients for Black-Box Variational Inference. ICML 2021
>
> [7] Szegedy, C., Vanhoucke, V., Ioffe, S., Shlens, J., and Wojna, Z. Rethinking the inception architecture for computer vision. CVPR 2015
>
> [8] Cao, H., Cooper, D. G., Keutmann, M. K., Gur, R. C., Nenkova, A., and Verma, R. CREMA-D: Crowd-sourced emotional multimodal actors dataset. IEEE transactions on affective computing, 2014.

---

### Official Review · Reviewer_FFkU · 2023-07-05

**Soundness:** 2 fair
**Presentation:** 3 good
**Contribution:** 2 fair
**Rating:** 5
**Confidence:** 4

**Summary:**

This work presents a novel implicit optimisation approach for Structured Variational Autoencoders (SVAEs) allowing for the computation of natural gradients in an automatic way. This allows for both an efficient learning scheme that is robust and avoids the bias found in other optimisation approaches of models that comprise Discrete variables, e.g., continuous relaxations. Experimental evaluations of these new optimisation schemes lead to SVAEs with competitive performance when compared to SOTA time series models while resulting in interpretable learned representations.

**Strengths:**

SVAEs constitute a powerful generative approach with potentially rich latent representations. The authors propose a new optimisation scheme that could bypass their training complexity and improve research explorations.

**Weaknesses:**

This work misses comparison against other related methods and some of the claims should be further clarified.

1) The continuous relaxation of the discrete variables do indeed produced a biased approximation, but have been shown to produce good
     results in practice, even without annealing schedules (line 117). For example see [3,4,5]. These claims should be validated with
     appropriate experimental evaluations in the considered setting.

2) Since the SVAE framework resorts to maximisation of a surrogate ELBO, doesn't it introduce a different bias? the same applies to when using the Richardson iteration; this approximation does introduce additional bias in the estimation, especially considering the capped implicit gradient estimator that the authors propose.

3) The related work is a bit lacking. The authors should discuss [2,6] and potentially compare against [6]. At the same time, there exists a newly published work by the same authors [7] that was accepted at ICML 2023, published before this submission. Additional discussions and comparisons should be included.

4) In this context, in my view there should be a comparison with other optimisation schemes for discrete latent variables such as the Gumbel-Softmax [1] or the NES strategy [2]. Even though the framework improves over the baseline and performs many times better than the competition, it is important to quantify the impact of the optimisation compared to more recent approaches than the biased estimation of the original SVAE publication.

5) What is the computational complexity of the framework in the various settings that the authors discuss?

[1] Maddison, C. J., Mnih, A., and Teh, Y. W. The concrete distribution: A continuous relaxation of discrete random variables, ICLR 2017

[2] Alon Berliner, Guy Rotman,Yossi Adi et al. Learning Discrete Structured Variational Autoencoder Using Natural Evolution Strategies,
ICLR 2022.

[3] KP Panousis, S Chatzis, S Theodoridis, Nonparametric Bayesian Deep Networks with Local Competition, ICML 2019

[4] K Kalais, S Chatzis, Stochastic deep networks with linear competing units for model-agnostic meta-learning, ICML 2022

[5] Sebastian Jaszczur, Aakanksha Chowdhery, Afroz Mohiuddin, et al., Sparse is Enough in Scaling Transformers, NIPS 2021

[6] Yixiu Zhao, Scott W. Linderman, Revisiting the Structured Variational Autoencoder, NIPS BDL Workshop 2021

[7] Yixiu Zhao, Scott W. Linderman, Revisiting Structured Variational Autoencoders, ICML 2023


**Questions:**

Please see the "Weaknesses" section.

**Limitations:**

Please see the "Weaknesses" section.

---

> ### Author Rebuttal · Authors · 2023-08-10
>
> Thank you for the added citations and comments. We will respond to each comment separately.
>
> 1: Thank you for the citations; added discussion of this model family is appropriate. For a discussion of these methods as an experimental baseline, please see the global response above.
>
> 2: The surrogate ELBO is used only to produce a variational posterior $q(z|x)$. Because we use a surrogate loss (and not the true loss), the posterior will be suboptimal. However, in amortized variational inference the posterior is always suboptimal! Thus the penalty for using a surrogate loss is simply an amortization gap, which is unavoidable in VAE learning. Unlike continuous relaxation-based methods, which relax the overall loss function and therefore perform descent on the wrong objective, all model parameters in the SVAE are optimized with respect to the true loss function.
>
> The Richardson iteration, if run to convergence, can exactly compute the implicit gradient. By capping the number of iterations we do introduce the possibility of biased gradients. This yields a tradeoff between compute and accuracy, unlike relaxation-based approaches which are biased even in the large-compute limit.
>
> 3: Thank you for the citations. We will add discussion of these papers to our related work. We would like to highlight that [7] was not published before our submission: it appeared on ArXiv on May 25th, a week after the NeurIPS submission deadline, and was published in the proceedings of ICML after that. We will add discussion of this work to section 7, and note that the paper differs from ours in 3 key respects. First, they show no experimental results on real data, only exploring the capacity of the SVAE to model synthetic and toy data. Second, because their experiments are restricted to the LDS graphical model (which requires no mean field factorization nor block updating) they do not need implicit differentiation, and do not explore the capacity of the SVAE to include discrete latent variables. Third, because they optimize point estimates of parameters instead of variational factors, they do not make use of natural gradients. In their point-estimate formulation, they can apply the parallel Kalman smoother of [1] off-the-shelf, whereas we derive a novel extension for our variational setting. Early experiments of ours revealed that using point estimates (and no natural gradients) barely hurts optimization of the LDS, but causes gradual pruning of discrete states in the SLDS until it has collapsed to a single mode. As this is the only difference between our implementations not already explored in our paper’s experiments, we would be happy to add results demonstrating this to our appendix in future revisions.
>
> Regarding [2], we would like to highlight that although it refers to “discrete structured VAEs”, these are unrelated to the SVAEs discussed in our paper and in [6-7]. [2] uses the concrete/gumbell softmax trick to model discrete latent variables.
>
> 4: Please see our global response for discussion of Gumbel-Softmax relaxations. We further highlight that no related work (to our knowledge) has provided a formulation for applying either scheme to dynamical VAEs for time series models.
>
> 5: Let D be the dimension of the continuous latent variable, and K be the dimension of the discrete latent variable. With a sequence of length T and L block updates, we have $O(LT(K^2+D^2 + KD))$ time complexity. Note that the $K^2 + KD$ factors can be parallelized to $\log K + \log D$ (they are matrix multiplies/element wise products followed by a sum along one axis), and our Kalman smoother can reduce the dependence on T to logT. Thus with optimal parallelization we have complexity $O(L \cdot \log T \cdot (D^2 + \log K))$. Because we do not store intermediate states our memory complexity is independent of L, capping out at $O(T(D^2+K^2))$.
>
> [1] Särkkä, S. and García-Fernández, Á. F. Temporal parallelization of bayesian smoothers. IEEE Transactions on Automatic Control, 2020
>
> [2, 6, 7] (we re-used your numbering to improved clarity of our response)

---

### Official Review · Reviewer_4krc · 2023-07-07

**Soundness:** 3 good
**Presentation:** 3 good
**Contribution:** 3 good
**Rating:** 7
**Confidence:** 4

**Summary:**

The paper introduces a novel, efficient learning algorithm for structured variational autoencoder for any graphical model.
It uses a block updating algorithm with belief propagation to approximate the posterior and employs implicit differentiation for end-to-end learning that is more memory-efficient than an unrolled alternative.
On top of that it derives unbiased natural gradients to improve the convergence of the learning and a few practical tricks such as parallel inference and robust initialization to make the implementation scalable.
Overall, the proposed algorithm makes SLDS models competitive to state-of-the-art time series models.


**Strengths:**

### originality
The proposed learning algorithm is novel.
Although each component seem to be well-established, combining them together is non-trivial.

### quality
The paper is solid and the method is sound.

### clarity
The paper is well-written and easy to follow.

### significance
The proposed method makes learning and inference of a wide range of graphical models practical through SVAEs, which is very important to the filed of probabilistic ML.

**Weaknesses:**

### quality

I think the experiments of the paper can be improved to incorporate more graphical models.
For now, the paper only tests the learning method on SLDS even though the method can be applied to a wide range of graphical models.

### clarity

The arrows for inference and generative networks in Figure 2 are pretty small.

**Questions:**

If the straight-through estimator is used, is the natural gradient still unbiased as a whole?

Are the time reported in Table the overall time of the 3-stage training or the number without the initialization (which seems to be expansive)?

**Limitations:**

Limitations are not discussed in the paper.
Some of my questions above could be discussed in a dedicated limitation section/paragraph.

---

> ### Author Rebuttal · Authors · 2023-08-10
>
> Thank you for your compliments and questions. We will aim to address your questions and concerns individually.
>
> Despite the SVAE’s flexibility to handle arbitrary graphical models–one of its most exciting properties, in our opinion–we focused on time series models due to space limitations, and with the goal of providing controlled comparisons to existing models on a standardized benchmark. We believe exploration of the model on novel tasks is important future work.
>
> Regarding Figure 2, we will resize and make better use of whitespace to improve clarity of the visualization.
>
> Regarding the straight-through gradient estimator, unlike prior works which make use of the estimator when gradients are intractable, in this case we are using it as a means of removing a Jacobian from the gradient computation. To compute the natural gradient (which is a preconditioned gradient and provably always a descent direction) we need to either use the straight-through gradient estimator or manually solve a system of linear equations induced by the Fisher Information matrix, which would prove much more expensive. Our resulting estimator of the natural gradient is unbiased.
>
> Table 3 does not include pre-training, but this initialization scheme is not expensive: because the standard VAE does not take into account time dependence (and therefore does not need to perform Kalman smoothing inference or RNN iteration) it is faster than any model discussed in the paper. Further, because models are trained for 200/100 epochs on Mocap/WSJ0 and we only pre-train for 10 epochs, the entire pre-training took <20 minutes on each data set.

---

### Official Review · Reviewer_JW8q · 2023-07-10

**Soundness:** 3 good
**Presentation:** 3 good
**Contribution:** 3 good
**Rating:** 7
**Confidence:** 3

**Summary:**

The paper builds upon a previous work (Structured VAEs) which tries to integrate variational auto encoders with probabilistic graphical models resulting in more expressing latent variable models for data. The authors propose a new unbiased estimator of the gradients for optimization the ELBO (for SVAE) and show via experiments that the superior performance of their algorithm. In particular, they show their structured model outperforms transformers and RNN based models on sequence prediction tasks (time-series predictions).

**Strengths:**

This paper makes learning SVAEs much more feasible by making the training more efficient due to the unbiased natural gradient estimators proposed by the authors.

The authors further propose an interesting implicit differentiation method to bypass the need to back propagate through the block coordinate ascent iterations while optimizing q(z) in the inner optimization loop. This would result in tremendous memory gains.

The paper and the appendix is well-written for most part (see weaknesses).

**Weaknesses:**

1. The paper is very clear as to its optimization innovations for improving the training to SVAEs but unclear or ambiguous in other aspects. In particular, one of the claimed contributions of this work is the ability to learn discrete latents (it is in the title of the paper). However, only one small section (S3.2) is devoted to this and the contribution is not clear. The way it is written, the solution seems to be simply removing the discrete variables from input to the decoder/inference network. It is not clear why this is not needed in SVAEs? What approximation is being made?

2. The authors claim that continuous relaxations of discrete distributions result in optimization procedures sensitive to the annealing schedule whereas their method is free from any such limitation since they do not need to backpropagate through discrete latents. I think an empirical comparison with VAEs that employ such relaxations like the Concrete VAE and the more popular VQ-VAE (this variant does not use a smooth relaxation of the discrete distribution but instead uses a straight-through estimator) would further strengthen this point. Without a comparison with contemporary methods it is hard to appreciate the benefit of SVAEs over these methods for optimizing over discrete latents.

**Questions:**

See the points made in the weaknesses. If my concerns are adequately addressed in the rebuttal I would be happy to improve my score. Here are few other minor concerns that could improve the readability of the paper.

1. I am not sure I understand the distinction between the true loss and the surrogate loss. Is the point that while optimizing q(z) (the latent variables) we learn an additional inference network \hat{l}(z | x) which parametrizes p(x | z) as an exponential distribution which is a conjugate to p(z)? Is \hat{l}(z | x) replaced by p_{\gamma}(x | z) while optimizing the outer loop of the ELBO with the locally optimal q(z)? Moreover, how is it ensured that \hat{l}(z | x) will be a good approximation of p_{\gamma}(x | z).

2. In Fig.2 and Sec 3.1 what are the disentangled factors and residual edges?

3. Lines 166-177 the capped implicit gradient estimator is not clear to me. If the Richardson iteration is run every step of the forward optimization. Does it mean that in every iteration of the inner loop when the \omega gets updated (by block coordinate updates), the gradient in eq (8) gets computed and the parameter \eta gets updated?

---

> ### Author Rebuttal · Authors · 2023-08-10
>
> Thank you for your compliments and your feedback. We will aim to respond to each comment individually.
>
> 1: The capacity of the SVAE to model discrete latent variables (without relaxation or bias) is a property of the SVAE model family, although it was not emphasized in the original paper which proposed the model. Our primary contribution is in improving and modernizing the SVAE so it can be tractably learned, but we argue it is a model family worth improving because of its discrete capabilities, which we demonstrate experimentally.
>
> Regarding the property itself, no approximation (other than the mean field approximation necessary to make inference tractable) is being used to allow for discrete variables; we simply need to specify our model such that the discrete latent variables are not direct inputs to the decoder network. For example, in the SLDS an encoder outputs variational factors on the continuous latent variables (Fig. 2), followed by block updating which estimates the posterior on both the continuous and discrete latent variables. This block updating causes the discrete distribution to impact the continuous variable and therefore the reconstructions, as the posterior on the continuous latent variables approximately marginalizes out the discrete switching states. We then sample continuous variables and decode. If in the Fig. 2 SLDS generative model, we had added $k_t$ as an input to the decoder network, we would have to sample $k$ and be in the standard VAE conundrum.
>
> We agree that added discussion of this matter would improve the paper and plan to expand section 3.2, including in ways that address your following comment.
>
> 2: For comments on the concrete VAE, please see the global response above. We agree that added discussion of the VQ-VAE is important, and plan to add it to discussion of related work in future revisions of the paper. Regarding experimental comparison, to our knowledge no one has successfully integrated either method into dynamical VAEs.
>
> Q1: The surrogate loss is only used to optimize $q(z)$, and the correct loss including $p_{\gamma}(x|z)$ is used in the outer loop. Because the ELBO is a valid lower bound for any choice of $q(z)$, any routine which produces a $q(z)$ induces a trainable family of models. Therefore, the SVAE’s choice of encoding data to variational factors and optimizing a surrogate loss is simply a reparameterization of $q(z|x)$ compared to the standard amortized setup where an encoder outputs the distribution parameters directly. It is not clear a priori which method which have a smaller amortization gap, but the desirable property of the SVAE is parameter sharing between the generative and inference model, because surrogate optimization at inference time uses $E_{q(\theta)}p(z|\theta)$.
>
> When $q(z)$ is true loss-optimal, our ELBO becomes a tight bound and our loss improves. In the fully-amortized setup, an encoder with sufficient capacity will eventually output a good estimator of q(z) because doing so will produce the best loss. In the SVAE, $q(z)$ is true loss-optimal when the true loss and surrogate loss look very similar, which in turn only happens when the encoder network outputs a good approximation of $p_{\gamma}(x|z)$. Just as training guides a standard encoder network to well-approximate the posterior, it guides the SVAE encoder to well-approximate the likelihood.
>
> Q2: In the true SLDS generative model, there is a joint distribution $p(z,k|\theta)$. However, for tractable inference we are learning a factorized posterior $q(z) q(k)$. The residual edges, shown in red, depict the dependencies that have been removed (i.e. which connect continuous and discrete latent variables) in this factorization, which are replaced by disentangled factors shown in blue.
>
> Q3: We perform L block update steps to compute omega (as part of the forward pass), followed by up to L Richardson iteration steps to compute the gradient of the loss with respect to eta (as part of the backward pass). Although the number of forward and backward steps are the same, they are not interwoven, and eta is only updated after the entire loss computation and backpropagation is complete.

---

> > ### Comment · Reviewer_JW8q · 2023-08-16
> > **Further Clarification**
> >
> > Thank you for your detailed responses. I had some follow-up questions.
> >
> > 1. The author's response does not answer my question as to how discrete variables are optimized in SVAEs. Could the authors illustrate their point with the simple example of the GMM L124-126?
> >
> > 2. The following line is unclear to me "we simply need to specify our model such that the discrete latent variables are not direct inputs to the decoder network." - Is this a modelling assumption or any given distribution involving discrete random variables can be written this way?
> >
> > 3. I am still not clear what the surrogate loss is. I understand it is still a lower bound, but what exactly is it? It will help to illustrate the surrogate loss vs true loss (which the VAE optimizes) using the standard normal example. This has been described in L81-84, but is not clear.

---

> > > ### Author Response · Authors · 2023-08-18
> > >
> > > Thank you for your questions! To answer **3**: Given a data point $x^{(i)}$, the ELBO-optimal posterior $q(z|x^{(i)})$ depends on both the prior $p(z|\theta)$ and the likelihood $p_\gamma(x^{(i)}|z)$. Because $x^{(i)}$ is a fixed value pulled from the data set, we need to think of the likelihood as a function over $z$, writing $\ell(z|x^{(i)})=p_\gamma(x^{(i)}|z)$.
> > >
> > > To find the optimal posterior, we must integrate $p(z|\theta)$ (which is easy) and $\ell(z|x^{(i)})$ (which is impossible due to the neural network decoder). The central innovation of the SVAE is that, instead of approximating the posterior directly as in amortized VI, we approximate the likelihood function $\ell(z|x^{(i)})$ and then "combine" it with the prior. By combine, we mean: we imagine a world where our prior is still $p(z|\theta)$ but our likelihood is now $\hat{\ell}_\phi(z|x^{(i)})$, and optimize the posterior in that setting. This is the surrogate objective: the ELBO in this hypothetical world.
> > >
> > > Let's consider the standard normal case with latent dimension 1. The prior is a $\mathcal{N}(0,1)$. Because we cannot integrate the decoder likelihood $\ell(z|x^{(i)})$ over $z$, we approximate it with a normal distribution whose mean is $\mu$ and precision (i.e. $\frac{1}{\sigma^2}$) is $\tau$. We write $\hat{\ell}_\phi(z|x^{(i)}) = \mathcal{N}(z;\mu,\tau^{-1})$, where $\mu$ and $\tau$ depend on $x^{(i)}$. In this setting, the surrogate loss is the ELBO given our $\mathcal{N}(0,1)$ prior and our $\mathcal{N}(\mu,\tau^{-1})$ (surrogate) likelihood. If the prior and likelihood are both Gaussian, then the optimal posterior (which minimizes the surrogate loss) is also Gaussian, with mean $\frac{\tau}{\tau+1}\mu$ and precision $\tau+1$. Note that in training we would never have to actually evaluate the surrogate loss: it is set up so we can find its optimizer in closed form.
> > >
> > > We now proceed to the Gaussian mixture model, where we will assume all model parameters ($\pi,\mu,\Sigma$) are point estimates for simplicity. The *full* generative model for an SVAE with a latent Gaussian mixture (and fixed output variances) is:
> > >
> > > $$k\sim\text{Cat}(\pi),\quad z \sim\mathcal{N}(\mu_k,\Sigma_k),\quad x\sim\mathcal{N}(\mu_\gamma(z),I)$$
> > >
> > > Where $\mu_\gamma(z)$ is the *decoder* network. The ELBO for this model is:
> > >
> > > $$\mathcal{L}= E_{q(k, z)}\left[ \log \frac{p(k)p(z\mid k) p_\gamma(x \mid z)}{q(k, z)}\right]\leq \log p(x)$$
> > >
> > > The amortized inference approach would estimate $q(k,z)$ via $q_\phi(k,z\mid x)$. However, in this case computing the loss requires sampling from $q_\phi(k, z\mid x)$, which is problematic with respect to gradients as discussed elsewhere (enumeration over $k$ is possible in the GMM case, but is infeasible for time-series models).
> > >
> > > An alternative used by several models is to make the (structured) mean-field assumption and assume that $q(k, z)$ factorizes as $q(k, z)=q(k)q(z)$.
> > >
> > > $$\mathcal{L}=E_{q(k)q(z)}\left[ \log\frac{p(k)p(z\mid k)p_\gamma(x \mid z)}{q(k)q( z)}\right]\leq\log p(x)$$
> > >
> > > The advantage of this factorization is that both $E_{q(k)}\left[\log p(k) \right]$ and $E_{q(k)q(z)}\left[\log p(z\mid k) \right]$ can be computed in closed-form and thus don't require sampling, while the reconstruction term: $E_{q(z)}\left[\log p_\gamma(x\mid z) \right]$ *no longer depends on $k$* and thus can be estimated with the standard reparameterization trick. As previously discussed, if the decoder took $k$ as input we would be left with $E_{q(z)q(k)}\left[\log p_\gamma(x\mid z, k) \right]$ and would be back to needing to sample $k$, thus to answer **2**, this is how we avoid sampling $k$ and it can be seen as a modeling choice. In the case of GMM, this choice does not hurt us: the continuous latent variable contains strictly more information than its cluster assignment so should be sufficient as an input to the decoder.
> > >
> > > If we again use amortized inference for both of these distributions, so that $q(k)$ becomes $q_\phi(k\mid x)$ and $q(z)$ becomes $q_\phi(z\mid x)$, we get the *stochastic inference network* (SIN) approach. This approach performs poorly in empirical tests due to the strong decoupling of $q(k)$ and $q(x)$ ($q(k)$ is also not directly informed by the decoder).
> > >
> > > The SVAE approach is to consider a world where the likelihood function isn't parameterized by neural networks, but is instead a simple Gaussian (the likelihood is only a function of $z$, not of $k$, so we don't need to define an approximate likelihood on $k$). GMMs with Gaussian likelihoods are well studied: in this case, optimizing the surrogate loss cannot be done in close form but can be done via *coordinate ascent variational inference* (CAVI). To answer **1**, CAVI alternates updates of $q(z)$ and $q(k)$ to optimize both using expectations, not samples. As a result, $q(k)$ can be optimized without sampling and its contributions to the loss can be evaluated without sampling, but it informs reconstruction: the parameters of $q(z)$ depend on the expectations of $q(k)$.

---

> > > > ### Comment · Reviewer_JW8q · 2023-08-21
> > > > **Update score**
> > > >
> > > > Thank you for the response. It addresses my concerns. Please add these details to the appendix since SVAE (in my opinion) is a niche subject and since the original paper does not answer some of these points well (like. ability to model discrete variables), I urge the authors to add this information in the appendix at thee very least.

---

### Author Rebuttal · Authors · 2023-08-10

We appreciate the feedback provided by all reviewers. In addition to the individual responses, we would like to highlight a point brought up by multiple reviewers: that we emphasize our model’s unbiased modeling of discrete latent variables but do not compare to the biased approach of the concrete (Gumbel-Softmax) VAE [1, 2].

To our knowledge, no one has successfully integrated concrete distributions into the model family discussed in this paper, dynamical VAEs. Thus, to make a comparison possible we have devised a new model which incorporates concrete distributions into a dynamical architecture, which we will call the “deep SLDS” (DSLDS) model. Results of this new comparison are provided below/attached.

The DSLDS has an identical generative model with the SLDS SVAE: switching linear dynamics in the latent space followed by a neural network decoder. However, instead of the SVAE surrogate loss trick, its inference is fully amortized and closely resembles inference in the DKS: a network independently encodes the observation at every time step, which are then used to iteratively sample $z_t$, $k_t$ in a Markovian fashion (i.e. k_t is sampled  given $k_{t-1}, z_{t-1}$ as well as $x_t$, and $z_t$ is sampled given $k_t, z_{t-1}$, and $x_t$). An intuitive visualization of the inference model is included in the attached figures. By combining the inference parameterization of the DKS and the generative parameterization of the SVAE SLDS, we can make a fair comparison between the two approaches.

Our experiments produce a model with reasonable reconstructions but very poor generations (see attached pdf). The deep SLDS learns latent dynamics that quickly diverge, leading to unrecognizable generations. For comparison with the results in table 2: the estimated likelihood ($> \log p(x)$) for the DSLDS is **1.953** for Mocap and **1.278** for WSJ0. The resulting generations produced “inception” activations far from those of real data; attempting to estimate FID scores with these generations resulted in numerical issues (the numbers would be far larger than those for competing methods).

As VAEs do not generate at training time, a large amortization gap can lead to this kind of gap in generative and reconstruction performance. The SVAE resolves this issue by using generative parameters at inference time; this technique cannot be easily applied to models whose latent variables follow a concrete distribution because it is not an exponential family. Approximation errors may be magnified in dynamical VAEs, because small differences in parameters can compound over time to produce large changes in generations at later time steps.

There are other reasons to prefer the SVAE setup to concrete relaxations: [3] demonstrated that natural gradient descent outperforms concrete approximations for learning parameters of discrete distributions. In addition, the original concrete paper [1] focused on binary variables and noted difficulty scaling to discrete variables with a higher number of states (they demonstrate this by changing the variable arity from 4 to 8, whereas we use 50). They note that for K discrete states, the temperature must be less than 1/(1-K) to sufficiently approximate the discrete distribution, but also note that as K increases, higher temperatures are necessary to fight the increased peakiness of the concrete distribution. In our DSLDS experiments, we were only able to set the temperature as low as 0.2-0.3 before encountering numerical issues. We believe our results reflect the difficulty of the concrete distributions to handle 50-ary discrete random variables.

[1] Maddison, C. J., Mnih, A., and Teh, Y. W. The concrete distribution: A continuous relaxation of discrete random variables. ICML 2017.

[2] Eric Jang, Shixiang Gu, and Ben Poole. Categorical Reparametrization with Gumbel-Softmax. In International Conference on Learning Representations (ICLR 2017), 2017.

[3] Ji, G., Sujono, D., and Sudderth, E. Marginalized Stochastic Natural Gradients for Black-Box Variational Inference. ICML 2021

---

### Decision · Program_Chairs · 2023-09-21

**Decision:**

Accept (poster)

**Comment:**

The work adapts statistical properties of the exponential family and their natural representation to structural variational auto-encoders. The work ties together many components and with this, its readability can definitely be improved to have an impact to the community. Despite my excitement about this submission I would like to echo the discussion: "The primary purpose of a scientific paper is to communicate a novel idea and share advances made in the field. Simplifying an idea, method or experimental protocol is a difficult but necessary step to achieve this, and failure to do so would only hurt the long-term impact of the paper."